# IDH3γ functions as a redox switch regulating mitochondrial energy metabolism and contractility in the heart

Maithily S. Nanadikar [1], Ana M. Vergel Leon [1], Jia Guo [1], Gijsbert J. van Belle [1], Aline Jatho[1], Elvina S. Philip [1], Astrid F. Brandner [2,16], Rainer A. Böckmann [2,3], Runzhu Shi [1], Anke Zieseniss[1], Carla M. Siemssen[1], Katja Dettmer [4], Susanne Brodesser [5], Marlen Schmidtendorf [5], Jingyun Lee[6], Hanzhi Wu[6], Cristina M. Furdui [6], Sören Brandenburg [7,8], Joseph R. Burgoyne [9], Ivan Bogeski[1], Jan Riemer[10], Arpita Chowdhury [11], Peter Rehling [11,12,13,14], Tobias Bruegmann [1,8,12], Vsevolod V. Belousov[1,15] & Dörthe M. Katschinski [1,8] ✉

Redox signaling and cardiac function are tightly linked. However, it is largely unknown which protein targets are affected by hydrogen peroxide ($H_2O_2$) in cardiomyocytes that underly impaired inotropic effects during oxidative stress. Here, we combine a chemogenetic mouse model (HyPer-DAO mice) and a redox-proteomics approach to identify redox sensitive proteins. Using the HyPer-DAO mice, we demonstrate that increased endogenous production of $H_2O_2$ in cardiomyocytes leads to a reversible impairment of cardiac contractility in vivo. Notably, we identify the γ-subunit of the TCA cycle enzyme isocitrate dehydrogenase (IDH)3 as a redox switch, linking its modification to altered mitochondrial metabolism. Using microsecond molecular dynamics simulations and experiments using cysteine-gene-edited cells reveal that IDH3γ Cys148 and 284 are critically involved in the $H_2O_2$-dependent regulation of IDH3 activity. Our findings provide an unexpected mechanism by which mitochondrial metabolism can be modulated through redox signaling processes.

The term oxidative stress refers to an imbalance between the production of reactive oxygen species (ROS) and endogenous antioxidant defense mechanisms. ROS consist of molecules that include reactive free oxygen radicals and the stable non-radical oxidant hydrogen peroxide ($H_2O_2$). $H_2O_2$ is considered to be a chief signaling molecule due to its longer half-life and capacity to diffuse through membranes[1]. An overshoot of $H_2O_2$ impairs cell function through oxidative damage of proteins, thereby, inducing cellular dysfunction or death[2]. The redox-modified molecules are mostly affected non-specifically and at random. In contrast to this detrimental function, there is strong evidence that $H_2O_2$ at low levels can act as specific signal transducer[3]. $H_2O_2$ exerts its effects by reversibly oxidizing protein cysteine thiols (-SH) to sulfenic acid (-SOH) that can finally result in a disulfide-bridge formation[4]. A protein cysteine thiol that can be reversibly oxidized by $H_2O_2$ is referred to as redox switch.

Elucidation of redox switches provides a better understanding of the role of redox signaling in health and disease. For the identification of specific redox switches, mass spectrometry (MS)-based redoxome analyses have gained importance[5–7]. These methods allow quantitative and site-specific redox-proteomic analyses. They have revealed many different bacterial and eukaryotic proteins, that use reversible thiol modifications to rapidly adjust their protein activity to the redox environment[8,9]. Although significantly advanced during the last years, there are still some drawbacks. In many cases studied, the concentrations of $H_2O_2$ are considerably larger than the concentrations found in vivo. In addition, for many of the proteins that were identified to be

redox-modified, the consequences of this post-translational modification are still unknown. To understand redox protein targets, redox modifications and their functional integration into the cellular physiology, defined and close to physiological conditions need to be applied. Genetically encoded redox sensors have been developed in the past to detect $H_2O_2$ levels with high sensitivity[10–12]. These fluorescent protein-based redox probes demonstrate high specificity and offer the possibility of transgenesis and subcellular targeting[13]. One prime example is the Hydrogen Peroxide sensor HyPer that allows real-time monitoring in living cells and tissues[14]. Complementary, chemogenetic tools like the D-amino acid oxidase (DAO) allow a precise and dynamic control of $H_2O_2$ levels[15]. Previous studies suggest that $H_2O_2$ is involved in the regulation of cardiac function[16]. However, a conclusive functional mechanism for this relationship is pending. Here, we generated HyPer-DAO transgenic mice that express the $H_2O_2$ generator and sensor specifically in cardiomyocytes. Phenotyping of the HyPer-DAO mice, together with redox-proteomics and cysteine gene editing in a cell model revealed that the γ-subunit of the isocitrate dehydrogenase (IDH)3 acts as a redox switch. Redox-modification of IDH3γ has profound impact on metabolic adaptation towards increased intracellular $H_2O_2$ levels and consequently cardiac function.

## Results

### Generation of cardiomyocyte-specific HyPer-DAO transgenic mice

The HyPer-DAO fusion protein allows to stimulate and analyze endogenous $H_2O_2$ production simultaneously[17]. To apply this tool in an in vivo setting, we generated mice that express the HyPer-DAO localized to the nucleus in cardiomyocytes (Fig. 1A). We have chosen the nucleus for the expression of the HyPer-DAO fusion protein to create an intracellular $H_2O_2$ gradient. Successful and tissue-specific expression of HyPer-DAO in the heart of the transgenic mice was verified (Fig. 1B–D). Cardiac function was unaltered in the resting HyPer-DAO mice compared to wild type (wt) siblings (Fig. 1E). The response of isolated cardiomyocytes towards the DAO-specific substrate D-ala was tested to verify the functionality of the HyPer-DAO fusion protein. Treatment of cardiomyocytes from HyPer-DAO mice with D-ala resulted in a dose-dependent $H_2O_2$ response (Fig. 1F). Cardiomyocytes did not respond to the enantiomer L-ala proving the specific function of the DAO.

### Intracellular $H_2O_2$ production in cardiomyocytes results in a reversibly impaired inotropic capacity

To understand the functional consequences of intracellular $H_2O_2$ production, HyPer-DAO mice were fed with D-ala or as a control L-ala in the drinking water. Seven and 21 days later, we analyzed the response on target RNAs of the redox-sensitive NF-κB and Nrf2 in the hearts of D-ala treated HyPer-DAO versus D-ala treated wt mice (Fig. 2A). NF-κB and Nrf2 regulate the immune and adaptive cellular responses to oxidative stress, respectively[18]. At low $H_2O_2$ levels, Nrf2 is induced and transactivates antioxidant enzymes, whereas an intermediate amount of $H_2O_2$ triggers the activation of NF-κB[19]. We observed a mild activation of the Nrf2 response, whereas the NF-κB response was almost absent except for an increase of iNOS RNA levels in mice after 21 days of D-ala treatment. After two additional treatment-free days subsequently to 21 days of D-ala treatment, the Nrf2 target genes were no longer increased. In line, we found just a minor increase in the RNA levels of redox-active enzymes excluding a massive activation of inflammation and adaptive stress response (Fig. 2B). We concluded that the mouse model allows us to study $H_2O_2$ as a second messenger in cardiomyocytes rather than as a mediator of a non-specific inflammatory oxidative stress response.

   We found striking effects of the intracellular $H_2O_2$ production on cardiac function. Ongoing treatment of HyPer-DAO mice with D-ala resulted in a significantly impaired cardiac pump function reflected by a decreased FAS, EF and SV in male and female mice, whereas left ventricular anterior and posterior wall thickness as well as LVIDs, LVIDd and EDV were unchanged (Fig. 2C and Supplementary Fig. 1A, B). To exclude a transgenic mouse-specific effect, HyPer-DAO mice were also treated with L-ala, which did not induce any significant changes in heart function. Likewise, wt mice did not show altered cardiac function after treatment with either D- or L-ala. Interestingly, the $H_2O_2$-induced impairment of heart function in the HyPer-DAO mice was reversible after only two D-ala treatment-free days (Fig. 2D). A subsequent 7 days long D-ala treatment again resulted in impaired heart function. After this second impairment, cardiac function still recovered when the D-ala treatment was suspended, although the recovery took longer compared to the response to the first deterioration. Intriguingly, even an impaired contractile function after a longer D-ala treatment, i.e. 21 days, was reversible after 2 days without D-ala (Supplementary Fig. 1C). In line with these findings, heart weight was unaltered in consequence of D-ala treatment (Fig. 2E), whereas ATP levels in left ventricular tissues tended to be lower after 7 days of D-ala treatment (Fig. 2F). Heart failure is usually relentlessly progressive as the maladaptive processes cause structural changes including fibrotic remodeling[20]. In line with the reversible phenotype, we did not detect a significant extent of fibrosis in the left ventricles (Fig. 2G and Supplementary Fig. 1D). As a positive control for the fibrotic staining, heart sections of mice that underwent transverse aortic constriction (TAC) were used. $H_2O_2$-induced impaired inotropic capacity was also evident by a D-ala-induced decrease of force generation in cardiac slices obtained from HyPer-DAO mice compared to wt mice (Fig. 2H and Supplementary Fig. 1E). The impairment was corrected by pre-incubation with N-acetyl-cysteine (NAC), an antioxidant and disulfide breaking agent[21], which indicates the marred contractility in the HyPer-DAO hearts was at least partially due to cysteine redox modifications.

### Redox proteome screen uncovers the mitochondrial IDH3γ as a $H_2O_2$ target

To identify specific redox-modified protein targets in the cardiac tissue of HyPer-DAO mice a proteomics approach was performed (Fig. 3A). We analyzed heart samples of 7 days D-ala-treated HyPer-DAO mice in comparison with wt mice. Successful redox enrichment was demonstrated by a significant number of unique peptides and total number of identified peptide spectra matches, which contain cysteines (Fig. 3B). In total we identified 6374 cysteine sites in 5391 unique peptides. Among those 185 peptides demonstrated a significant reduction of reversible oxidation and 115 peptides demonstrated a significant increase in their reversible oxidation (Fig. 3C and Supplementary Tables 1, 2). Based on the findings that the impaired heart function was reversible, we concentrated on modifications on proteins involved in mitochondrial metabolism. Strikingly, the TCA cycle enzymes succinate dehydrogenase (SDH) and isocitrate dehydrogenase (IDH)3 were identified in the screen. Aside from these, we identified Prx5. Prxs are a family of thiol peroxidases that scavenge $H_2O_2$ in cells[22], which demonstrates the validity of the screen performed.

   SDH is localized in the inner mitochondrial membrane and participates in both the TCA cycle and the electron transport chain (ETC). IDHs catalyze an oxidative decarboxylation of isocitrate to α-ketoglutarate[23]. IDH1 exerts its function in the cytosol and peroxisomes, whereas IDH2 and IDH3 both localize to the mitochondria. IDH1 and IDH2 are homodimers that share sequence similarity (Fig. 3D). They catalyze the oxidative decarboxylation of isocitrate to 2-ketoglutarate to generate NADPH from $NADP^+$ and the reverse reaction. By contrast, IDH3 is a hetero-tetramer, composed of two α, one β and one γ subunit, encoded by three independent genes. From

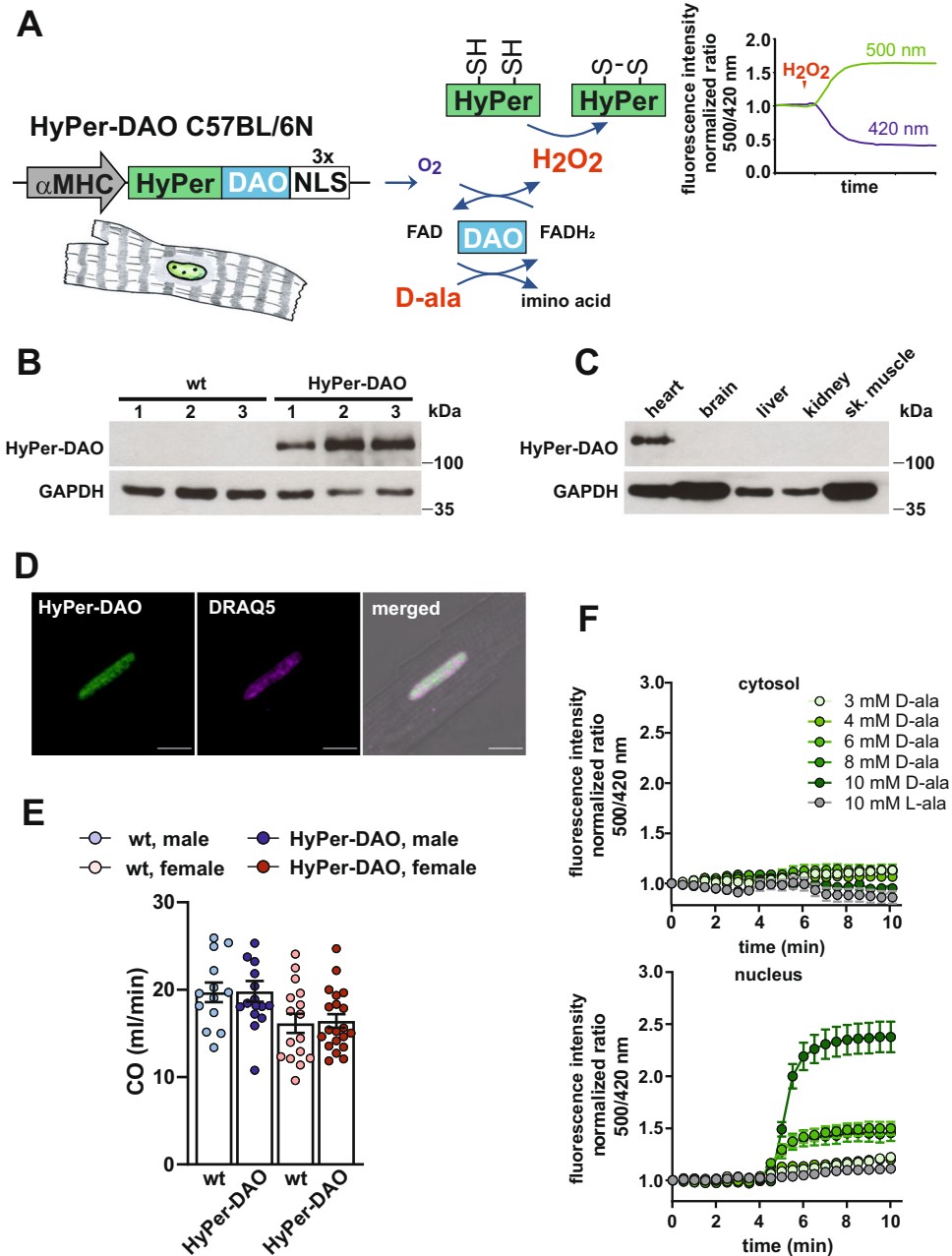

**Fig. 1 | Expression and activation of the HyPer-DAO fusion protein in cardiomyocyte-specific transgenic mice. A** Scheme of the HyPer-DAO construct used for generating HyPer-DAO cardiomyocyte-specific transgenic mice and the HyPer and DAO chemical reactions. **B** Immunoblot for HyPer-DAO and GAPDH protein levels in cardiac protein lysates from three independent wild type (wt) and three independent HyPer-DAO mice. **C** Cardiac-specific HyPer-DAO protein expression in a HyPer-DAO mouse as demonstrated by immunoblot performed with protein lysates from different organs. This confirmatory experiment has been performed once. **D** Representative confocal image of a DRAQ5 stained cardiomyocyte isolated from a HyPer-DAO mouse. In total cardiomyocytes from two independent mice were isolated and 5 cardiomyocytes per mouse were imaged. **E** Cardiac output (CO) as determined by echocardiography in resting male and

female HyPer-DAO and wt mice ($n$ = independent 13 wt male, 15 HyPer-DAO male, 16 wt female and 20 HyPer-DAO female mice). **F** Response of HyPer-DAO cardiomyocytes to treatment with D-ala or L-ala (added at time point 4 min). The HyPer fluorescence response was recorded in the cytosol (upper panel) and the nucleus (lower panel). Ratios are normalized to the HyPer ratio prior to treatment, $n$ = 27, 32, 47, 56, 38 and 13 independent cardiomyocytes were analyzed regarding their response in the nucleus and $n$ = 23, 26, 13, 29, 25, and 17 cardiomyocytes regarding their response in the cytoplasm were analyzed after treatment with 3 mM D-ala, 4 mM D-ala, 6 mM D-ala, 8 mM D-ala, 10 mM D-ala and 10 mM L-ala, respectively. scale bar, 10 μm. Data are presented as mean values ± SEM. Source data are provided as a Source Data file.

these, the γ-subunit was found in the redox-proteomic screen. The peptide containing IDH3γ Cys148 was successfully measured with high confidence (XCorr values ranging from 2.59-3.63). The fold change for Cys148 was found to be 1.52, $p$-value = 0.016. Unlike IDH1 and 2, IDH3 catalyzes the irreversible conversion of isocitrate to 2-ketoglutarate while reducing NAD$^+$ to NADH.

### IDH3 activity is regulated by a reversible redox modification in vitro and in vivo

Having found SDH as well as IDH3γ, we next analyzed the activity of TCA cycle enzymes in heart lysates of these mice. IDH3 activity was significantly increased in D-ala-treated HyPer-DAO compared to wt samples (Fig. 3E). This effect was specific to IDH3 since neither IDH1/2,

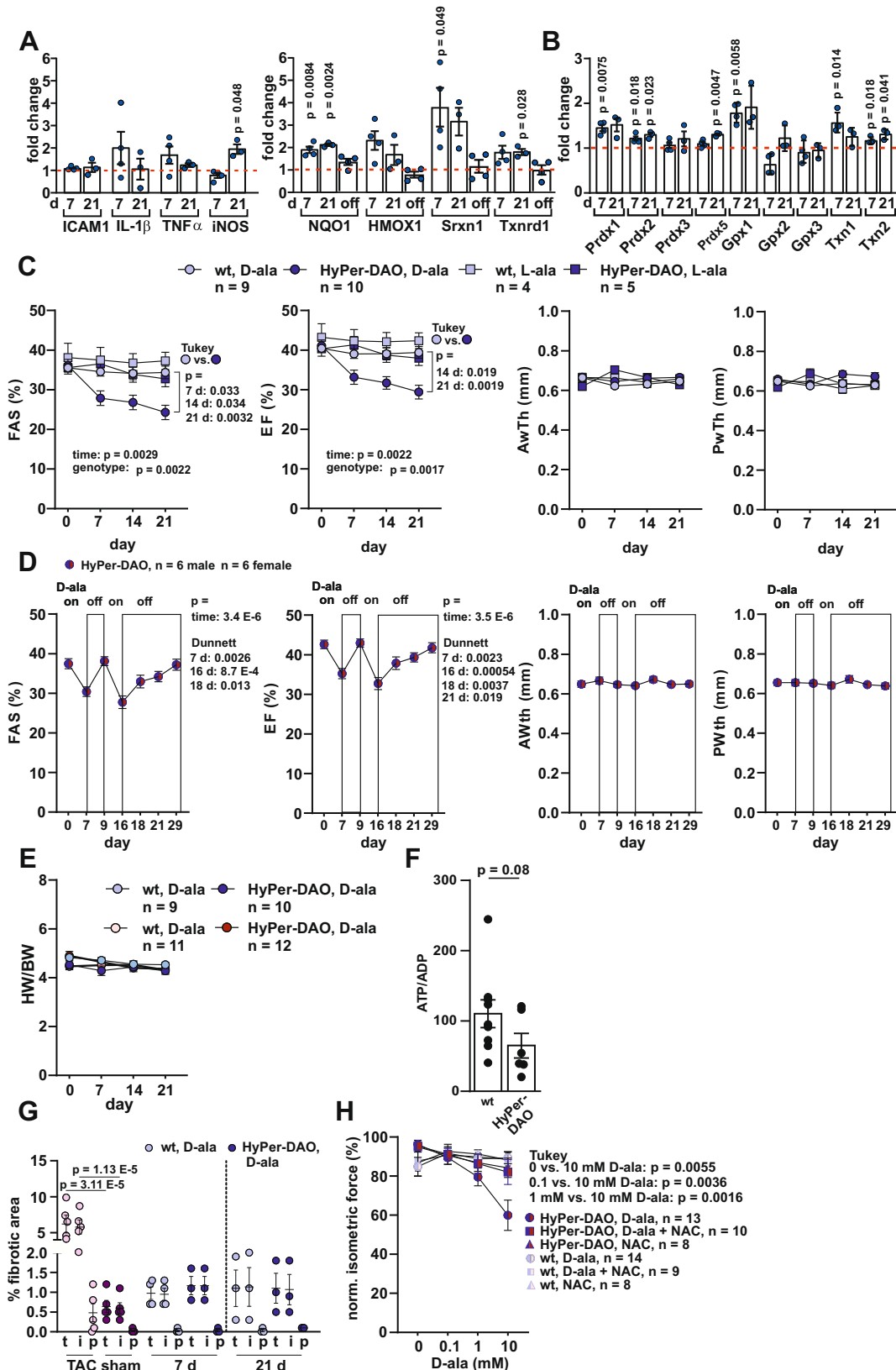

fumarase, malate dehydrogenase nor especially SDH were affected. Therefore, we concentrated in our subsequent studies on the identified redox-modification of IDH3γ. The increased IDH3 enzyme activity was not related to increased IDH3γ RNA (Fig. 3F) or protein levels (Fig. 3G). A reversible modification was further supported by the finding that in 7 days D-ala-treated mice IDH3 activity went back to

normal after feeding the animals for 2 days with D-ala free drinking water (Fig. 3H). Increased IDH3 activity was likewise detected in cardiomyocytes isolated from HyPer-DAO mice after D-ala treatment in vitro (Fig. 3I). ROS are associated to the development of pressure overload-induced heart failure[24]. We investigated, if IDH3 activity and IDH3γ modification are also altered in the hearts of respective mice

**Fig. 2 | Reversible impaired inotropic response of HyPer-DAO mice towards activation of DAO.** Relative changes in (**A**) RNA levels of NF-κB (left panel) and Nrf2 (right panel) target genes and (**B**) antioxidant enzymes in the heart of HyPer-DAO male mice versus wild type (wt) male mice after 7 days ($n = 4$ independent wt and $n = 4$ independent HyPer-DAO animals) and 21 days ($n = 3$ independent wt and $n = 3$ independent HyPer-DAO animals) of treatment with 55 mM D-ala in the drinking water. For Nrf2 target RNAs in addition mice that received D-ala for 7 days in the drinking water and subsequently 2 days treatment-free drinking water (off) were analyzed, $n = 4$ independent HyPer-DAO mice. **C**, **D** Echocardiographic analysis of fractional area shortening (FAS), ejection fraction (EF), anterior wall thickness (AWth) and posterior wall thickness (PWth) in male (in C) and a mixed group of female and male (in D) HyPer-DAO or wt mice after treatment with 55 mM D-ala or L-ala in the drinking water as indicated. **E** Heart weight to body weight (HW/BW) in female and male HyPer-DAO and wt mice before and after D-ala treatment. **F** ATP/ ADP levels in left ventricular samples obtained from $n = 9$ independent wt and $n = 6$ independent HyPer-DAO mice 7 days after D-ala treatment. **G** Quantification of Sirius red/Fast green staining for fibrotic areas in cardiac slices of male HyPer-DAO and wt mice. The total area of fibrosis (t), interstitial (i) and perivascular (p) fibrotic area were quantified in the left ventricles. Cardiac slices of mice that underwent transverse aortic constriction (TAC) or sham surgery were used as positive control for the staining and quantification. $n = 5$ independent TAC and sham mice, $n = 4$ (7 days) and $n = 3$ (21 days) independent wt mice and $n = 3$ (7 and 21 days) HyPer-DAO mice. **H** Heart slices isolated from a mixed group of female and male HyPer-DAO or wt mice were treated with 0-10 mM D-ala + /− 1 mM NAC and force development was measured. Data are presented as mean values ± SEM, two-tailed one sample $t$-test (**A**, **B**), one-way ANOVA (**D** and **G**), two-way ANOVA (**C** and **H**), and two-tailed unpaired $t$-test (**F**). This figure is related to Supplemental Fig. 1. Source data are provided as a Source Data file.

and found an increased IDH3 activity (Fig. 3J) as well as IDH3γ cysteine oxidation in left ventricular samples of mice 4 weeks post TAC (Fig. 3K).

To analyze IDH3 redox regulation at cellular level, we created HyPer-DAO HEK293 (HEK) cell models. We generated HyPer-DAO overexpressing HEK cells targeted to the nucleus, the mitochondria or the cytoplasm (HyPer-DAO-NLS, HyPer-DAO-MLS and HyPer-DAO-NES). These cells allowed us to address possible cellular compartment-specific effects, which we could not analyze in our mouse model. Correct localization of HyPer-DAO was confirmed by staining and confocal microscopy (Fig. 4A). All three cell lines responded to stimulation with increasing concentrations of D-ala but not L-ala with a typical HyPer response (Fig. 4B), which was similar to the response of the recycling drug menadione but not inhibitors of the ETC (Supplementary Fig. 2A). For further analysis we chose a D-ala concentration of 50 mM, which resulted in a comparable response in all three cell lines. To analyze how mitochondrial $H_2O_2$ levels are affected, we transiently overexpressed HyPerRed localized to the mitochondrial matrix in the HyPer-DAO-NLS, HyPer-DAO-MLS and HyPer-DAO-NES cells. Treatment of the cells with D-ala resulted in an increase of $H_2O_2$ levels in the mitochondria regardless of the original production site of $H_2O_2$ (Supplementary Fig. 2B). This finding is in line with earlier experiments demonstrating that $H_2O_2$ produced in the cytosol quickly reaches the mitochondrial matrix, whereas the opposite direction of transport by unknown mechanism is restricted[25,26].

Treatment of HyPer-DAO-NLS, HyPer-DAO-MLS and HyPer-DAO-NES HEK cells with D-ala for 20 min resulted in an increase in IDH3 activity in all three cell lines demonstrating that this effect is independent from the source of the increased $H_2O_2$ (Fig. 4C). An oxidation-dependent IDH3 activity was further supported by the fact that treatment of the cells with a bolus of $H_2O_2$ produced an effect comparable to D-ala treatment. In contrast, IDH1/2, SDH, fumarase or malate dehydrogenase activities were unchanged (Supplementary Fig. 3A). The D-ala-induced increased IDH3 activity was blunted by co-treatment with NAC. To exclude that NAC inhibits the amount of ROS produced in the cells rather than affecting protein redox-modifications, we determined the HyPer response and $O_2^-$ levels after D-ala treatment. Whereas the HyPer response, which relies on a reversible disulfide-bridge formation, was blunted by co-stimulation with D-ala and NAC, $O_2^-$ levels were unaffected (Fig. 4D). A cysteine oxidation in IDH3γ after D-ala or $H_2O_2$ treatment was verified in PEG switch assays in the HyPer-DAO-NLS, HyPer-DAO-MLS and HyPer-DAO-NES HEK cells further demonstrating that the effect can be observed independent from the primary production site of $H_2O_2$ (Fig. 4E and Supplementary Fig. 3B). We concentrated in the following experiments on the HyPer-DAO-NLS HEK cells. The D-ala-induced increase in IDH3 activity and IDH3γ cysteine oxidation was reversible upon washing off the D-ala (Fig. 4F, G). To exclude non-specific effects of D-ala, we treated HEK wt cells with D-ala, L-ala and $H_2O_2$, from which only $H_2O_2$ induced an increased IDH3 activity (Supplementary Fig. 3C). An $H_2O_2$ or D-ala

induced increased IDH3 activity was additionally confirmed in MDA-MB-231, Hep3B and C2C12 cells as well as in HyPer-DAO-NLS and HyPer-DAO-NES expressing Hep3B cells demonstrating that this effect is cell line independent (Supplementary 3D). The response towards $H_2O_2$ was seen at different concentrations in each cell line, which might be due to different anti-oxidant capacities of the cells.

## Stimulation of endogenous $H_2O_2$ production results in an impaired ATP production in mitochondria

To gain insight into the energy status of the HyPer-DAO cells, we analyzed the consequences of D-ala, L-ala and $H_2O_2$ treatment on the cellular ATP levels. ATP levels were significantly decreased in HyPer-DAO-NLS cells after treatment with D-ala or $H_2O_2$, whereas ATP levels dropped after $H_2O_2$ treatment in HEK wt cells only (Fig. 5A). This effect was partially prevented by co-treatment with NAC. Mitochondrial and glycolytic ATP synthesis were likewise impaired after D-ala treatment in HyPer-DAO-NLS cells (Fig. 5B). An impairment of mitochondrial ATP synthesis was also observed in cardiomyocytes isolated from HyPer-DAO mice after treatment with D-ala via analyzing $O_2$ consumption (Fig. 5C). Since the DAO reaction per se is $O_2$ dependent, the $O_2$ consumption rate analysis mostly reflects altered ETC activity. In addition, the ratio of reduced glutathione to oxidized glutathione (GSH/GSSG) was significantly decreased in D-ala treated HyPer-DAO-NLS cells (Fig. 5D), which was accompanied by a depletion of total glutathione levels (Fig. 5E).

## Cys148 and Cys284 are key for $H_2O_2$-induced changes in IDH3 activity

IDH3γ contains in total 6 cysteines from which only Cys148 was found to be significantly redox-modified in the redox-proteomic screen. IDH3γ Cys148 also appeared in a previously published database that was generated based on quantitative mapping of the in vivo mouse cysteine redox proteome[27] (Fig. 6A). To further gain insight into the importance of cysteine modification in IDH3γ, we applied an internet-based disulfide prediction tool[28]. A crystal structure of the α/γ heterodimer of human IDH3 in complex with $Mg^{2+}$, citrate and ADP were used as query protein structure to visualize an anticipated disulfide-bond[29]. The result revealed a close proximity between Cys148 and Cys284, which allows disulfide-bridge formation.

Cys284 was not identified in the redox-proteomic screen, which might rely on a lesser accessibility of this cysteine in a proteome-based screening. However, based on the strong prediction of the disulfide-bridge between Cys148 and Cys284, we went on to analyze the consequences of such posttranslational modification in IDH3γ. Human IDH3 is known to display enzymatic activity at different oligomerization levels, i.e. the hetero-dimer composed of IDH3α and IDH3γ subunits has basal activity, however the hetero-tetramer composed of the dimers IDH3α/γ and IDH3α/β has a much higher activity[30]. Furthermore, the IDH3α/γ dimer can be allosterically regulated by citrate and/or ADP[29]. As shown in Fig. 6B the allosteric site is located in the

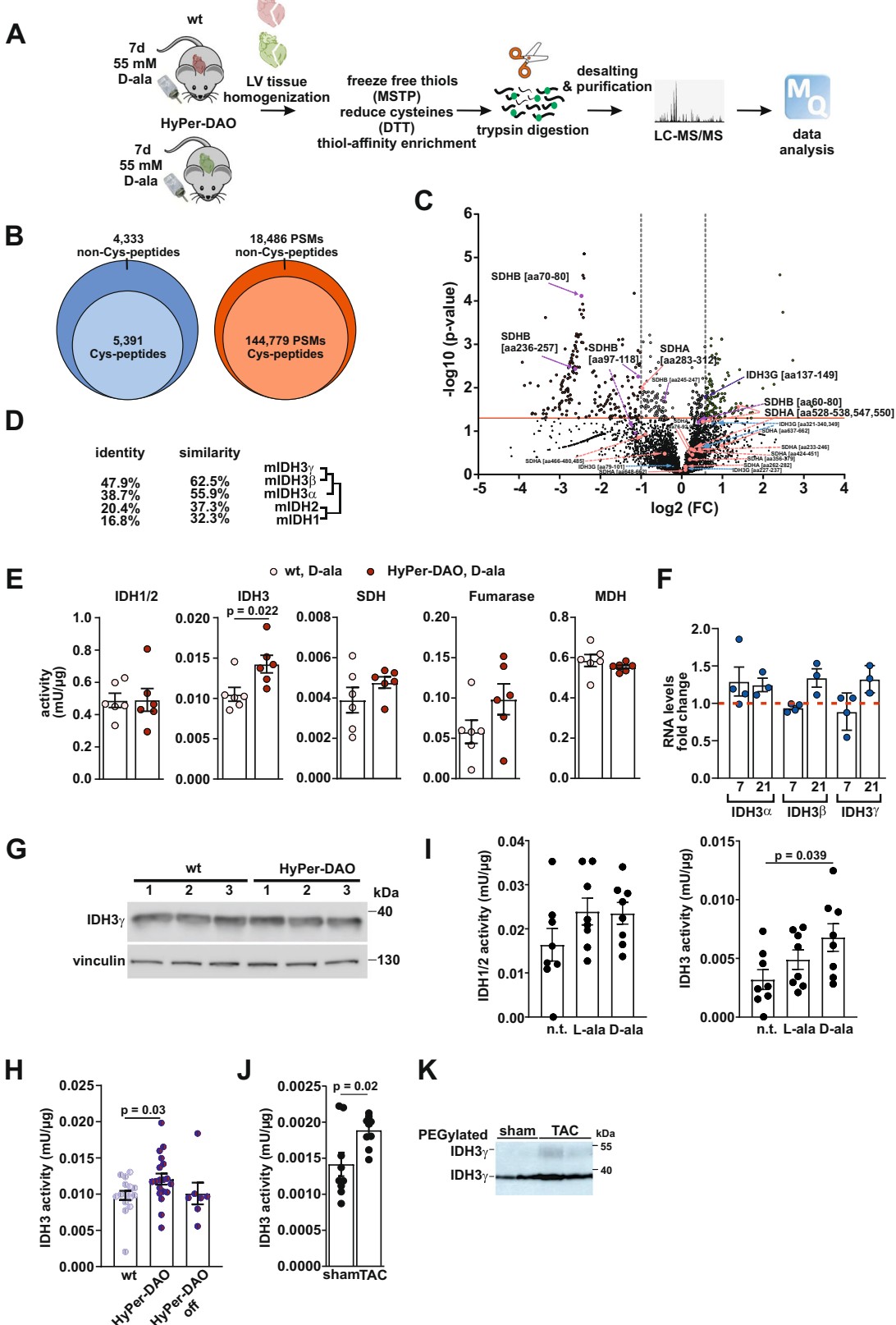

γ-subunit, whereas the catalytic site of the dimer is within the α-subunit. In addition, the clasp region is marked in the figure as it defines the tetramer interface between the dimers IDH3α/γ and IDH3 α/β. We ran 2 sets of 5 molecular dynamic simulations of the IDH3α/γ dimer in presence of $Mg^{2+}$, citrate, and ADP. Based on the evidence from the disulfide prediction, one set of simulations included a disulfide-bond between Cys284γ and Cys148γ (SS), whereas the other did not (noSS). A global root mean square deviation (RMSD) analysis of the simulations separately for the full dimer complex and for the individual subunits revealed that the systems without the disulfide-bond are marginally less stable, particularly regarding the α-subunit (Supplementary Fig. 4). More importantly, the RMSD of the clasp

**Fig. 3 | IDH3γ is redox modified in the hearts of HyPer-DAO mice after activation of DAO in vivo. A** Scheme of the redox-proteomics analysis performed with heart samples obtained from $n = 3$ independent male HyPer-DAO and $n = 3$ independent wt mice that were treated for 7 days with 55 mM D-ala in the drinking water. **B** Venn-diagrams demonstrate elevated yield of cysteine-containing unique peptides and total number of identified peptide spectra matches (PSM) by redox enrichment. **C** Volcano plot of the peptides identified in the redox-proteomics screen as outlined in **A**. fold change (FC) **D** Protein dendrogram for the different murine isocitrate dehydrogenase isoforms and subunits. **E** Activity of the indicated TCA cycle enzymes determined in protein extracts obtained from hearts of 7 days D-ala treated female HyPer-DAO and wt mice. $n = 6$ independent mice per group. **F** Relative changes in RNA levels of IDH3 subunits in the hearts of HyPer-DAO mice versus wt mice after 7 days ($n = 4$ independent wt and $n = 4$ independent HyPer-DAO animals) and 21 days ($n = 3$ independent wt and $n = 3$ independent HyPer-DAO animals) of treatment with D-ala. **G** Immunoblot for IDH3γ and vinculin protein levels in heart samples of three independent wt and three independent HyPer-DAO mice after 7 days of treatment with D-ala. **H** IDH3 activity in samples obtained from hearts of a mixed group of male and female HyPer-DAO or wt mice after 7 days treatment with D-ala ($n = 16$ independent wt and $n = 19$ independent HyPer-DAO animals) or after 7 days treatment with D-ala plus 2 additional D-ala free days (off, $n = 7$ independent HyPer-DAO animals). **I** IDH1/2 and IDH3 activity in protein samples obtained from isolated HyPer-DAO cardiomyocytes non-treated (n.t.) and after treatment with 2 mM D-ala or L-ala for 20 min. Cardiomyocytes were isolated from $n = 8$ independent HyPer-DAO mice. **J** IDH3 activity determined in protein extracts obtained from the left ventricular myocardium of mice 4 weeks after transverse aortic constriction (TAC) versus sham surgery. $n = 9$ independent mice per group. **K** Representative immunoblot for IDH3γ oxidation analyzed by PEG switch assay in left ventricular samples from TAC and sham treated mice. The experiment has been performed three times. Data are presented as mean values ± SEM, two-tailed unpaired $t$-test (**C**, **E** and **J**), two-tailed one-sample $t$-test (**F**), one-way ANOVA (**I**) and one-way Welch's ANOVA test (**H**). Source data are provided as a Source Data file.

domain of the IDH3α/γ dimer indicates that two of the noSS replicas are structurally diverging in that area (Fig. 6C); in particular the region spanning residues 149α to 163α, in the vicinity of the catalytically relevant Tyr153α, became unstructured (Fig. 6D). This conformational change particularly affects the tetramerization interface (depicted in blue) as compared to the reference simulations in presence of the disulfide-bond (depicted in cyan). In order to systematically identify the differences related to disulfide-bond we performed a force distribution analysis[31]. Figure 6E shows the residue-based stress difference (noSS-SS). The main residues with statistically significant changes in the stress profile are highlighted. Substantial differences are seen within the vicinity of Tyr153α with residues Glu150α and Glu152α (both at the beginning of the first β-sheet of the α subunit's clasp). Other residues were found close to the disulfide region such as Ile163γ (β-sheet in contact with S–S bridge, part of the rigid core of γ-subunit) and Tyr174γ (loop before clasp of γ-subunit) and Ile279γ which lies on the same α-helix as Cys284γ. A region with significant increase in the overall stress upon removal of the disulfide-bond is found at the apical domain of the γ-subunit, in vicinity of its termini (tan sticks in Fig. 6E). From a functional point of view, the N-terminal region of the γ-subunit has been shown to interact with the β-subunit cleft within the octameric conformation of IDH3[32]. The stress patterns overall suggest a stress transfer from the region of the disulfide-bond towards the clasp domain that might affect the tetramerization interface as it mainly implies changes on the β-sheets of the clasp domain that do not directly contribute to the dimerization interface but rather are part of the tetramerization interface.

### H₂O₂-dependent IDH3γ Cys148 modification affects TCA cycle turnover

To analyze the functional importance of the results obtained in the molecular simulation, IDH3γ Cys148 and Cys284 were gene edited to alanine in HyPer-DAO-NLS HEK cells (C148A and C284A). Basal IDH1/2 and IDH3 activities in the gene-edited cells were not significantly different compared to the wt cells (Fig. 7A, B). Treatment of the cells with D-ala however demonstrated that the H₂O₂-induced increase of IDH3 activity was abolished in the C148A and C284A cells. To exclude a non-specific effect of the cysteine mutation, we also gene edited IDH3γ Cys236 to alanine (C236A). The C236A cells had a trend towards lower IDH3 activity, however, responded to the D-ala treatment similar to the wt cells with an increased IDH3 activity. The non-responsive IDH3 activity in the C148A and C284A cells after D-ala treatment was accompanied by a loss of cysteine oxidation in the C148A but not the C284A mutant (Fig. 7C). Thus, IDH3γ Cys148 might act as the redox-active, peroxidatic cysteine, which is first oxidized and subsequently attacks Cys284 for disulfide binding. A lack of D-ala induced IDH3 activity was further confirmed in cells harboring mutations in both

critical cysteines (Supplementary Fig. 5A, B). We further went on to characterize the metabolism in the wt compared to the C148A cells. The abolished effect of H₂O₂-induced increase of IDH3 activity in the C148A cells was associated with a partial rescue of mitochondrial ATP production after D-ala treatment. This was seen by analyzing oxygen consumption rates as well as by employing a mitochondria-specific fluorescent ATP dye (Fig. 7D, E). Whereas both glycolytic and mitochondrial ATP levels dropped in the HyPer-DAO-NLS cells after D-ala treatment, the impairment in mitochondrial ATP was no longer observed in the C148A cells. Glycolytic ATP production, however, was still reduced in the Cys148 gene edited cells after D-ala treatment alike in the HyPer-DAO-NLS cells. This demonstrates that IDH3γ Cys148 and Cys284 are crucial for mediating the oxidation-induced mitochondrial metabolic changes but not the change in glycolytic ATP production. Rescued mitochondrial ATP levels were likewise seen in C284A and C148A/C284A (Supplementary Fig. 5C) but not in the C236A cells. Levels of TCA cycle intermediates upstream of the IDH3-catalyzed reaction, i.e. citrate/isocitrate were found to be decreased in D-ala induced HyPer-DAO-NLS cells compared to non-treated cells (Fig. 7F). A further significant decrease of the later intermediates (i.e., fumarate and malate) downstream of IDH3 in addition suggests a change in TCA cycle activity in this condition. This effect was partly reverted in the C148A mutated cells. The only metabolite that was found to be significantly increased in both the wt and the C148A cells was succinate and thus apparently unrelated to the IDH3γ C148 modification. Targeted stable isotope tracer studies using uniformly ¹³C-labeled glucose and GC-MS confirmed an increased TCA flux from citrate/isocitrate to α-ketoglutarate and succinate in the HyPer-DAO-NLS cells after D-ala treatment, which was not found in the C148A cells (Fig. 7G). Improved energetics in the C148A cells after D-ala treatment were also reflected by an increased NADH/NAD⁺ ratio compared to the wt cells associated with a rebalancing of reducing NADPH and GSH equivalents in HyPer-DAO-NLS and C148A cells (Fig. 7H, I). Overall, the data demonstrate that IDH3γ Cys148 oxidation under oxidative stress is altering mitochondrial ATP production by changes in TCA cycle flux (Fig. 7J).

### Discussion

H₂O₂ was long regarded as a non-specific damaging agent. This notion has shifted to recognizing its specific role in cell biology[33,34]. The conceptual change was in part triggered by the development of MS-based proteomic methods that reliably quantify protein oxidative modifications in cell and tissue samples[35] and revealed regulatory post-translational modifications. It is well recognized that alterations of the redox balance towards oxidation is associated with various pathological conditions, including cardiovascular diseases[36]. To gain a better understanding of specific protein targets affected by H₂O₂, the development of relevant cell and animal models is important. We used

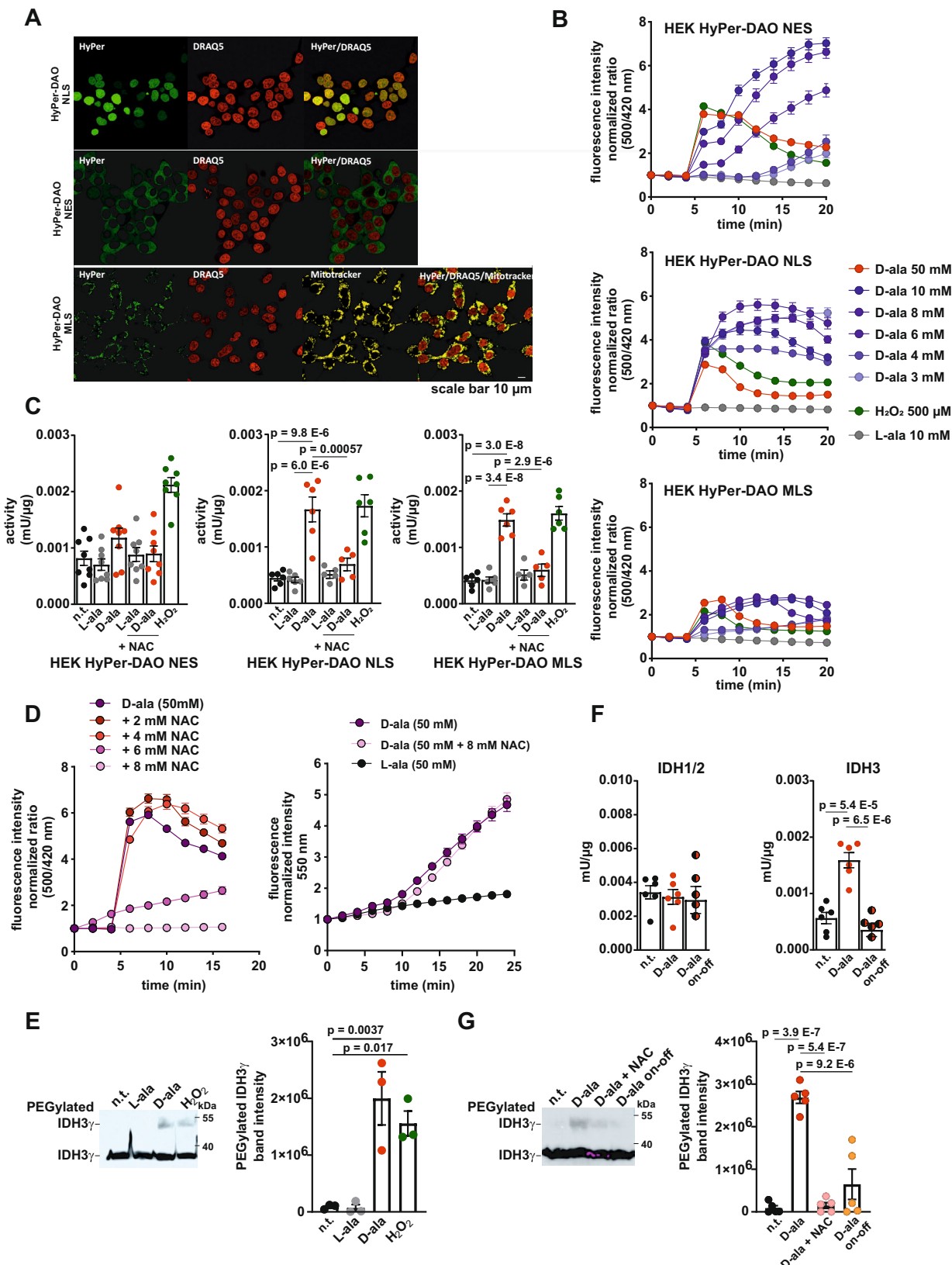

a newly generated chemogenetic mouse model in combination with MS-based redox-proteomics. A similar chemogenetic approach including DAO was applied before in rats, which were treated with a cardiotropic adeno-associated virus expressing HyPer-DAO under control of the cardiac-specific cTnT promoter[37]. The animals developed a cardiomyopathy and changes in the cardiac metabolome after D-ala treatment[38]. DAO stimulation induced a classical Nrf2 and Nf-κB response, which is of striking difference to our mouse model and might be due to a much longer (4–7 weeks compared to a maximum of 3 weeks in this study) treatment and an 18-fold higher D-ala concentration applied. Redox-target proteins were not reported in the rat model. Using tissues of the transgenic mice revealed the TCA cycle

**Fig. 4 | HyPer-DAO overexpressing HEK cells exhibit a reversible redox modification and activity of IDH3 after activation of DAO in vitro. A** Confocal images of HEK cells overexpressing the HyPer-DAO fusion protein in the nucleus (HyPer-DAO-NLS), the cytoplasm (HyPer-DAO-NES) or the mitochondrial matrix (HyPer-DAO-MLS) after the indicated staining. The staining has been performed once. **B** HyPer fluorescence response in HyPer-DAO overexpressing HEK cells to treatment with L-ala, D-ala or $H_2O_2$ (added at time point 4 min). Ratios are normalized to the HyPer ratio prior to the stimulation, $n = 30$ independent cells per condition were analyzed except than $n = 25$ independent HyPer-DAO-NES and HyPer-DAO-NLS cells treated with 500 µM $H_2O_2$, see also Supplemental Fig. 2. **C** IDH3 activity in cell extracts obtained from HEK HyPer-DAO-NES, HEK HyPer-DAO-NLS and HEK HyPer-DAO-MLS cells non-treated (n.t.) and after treatment with 50 mM D-ala, 50 mM L-ala ± 8 mM NAC or 500 µM $H_2O_2$ for 20 min. $n = 8$ independent experiments for HyPer-DAO-NES all conditions; $n = 6$ independent experiments for HyPer-DAO-NLS and HyPer-DAO-MLS all conditons except for $n = 5$ for L-ala and D-ala + NAC, see also Supplemental Fig. 3. **D** HyPer fluorescence (left panel) and MitoSOX (right panel) response in HyPer-DAO-NLS cells to treatment with L-ala, D-ala ± NAC (added

at time point 4 min). Ratios are normalized to the HyPer ratio or the MitoSOX fluorescence prior to the stimulation, $n = 30$ independent cells for all HyPer measurements and $n = 25$ independent cells for all MitoSox measurements. **E** Quantification of IDH3γ oxidation analyzed by PEG switch assay in HEK HyPer-DAO-NLS cells n.t. and after treatment with 50 mM L-ala, 50 mM D-ala or 500 µM $H_2O_2$ for 20 min. $n = 3$ independent experiments. A representative experiment is shown in the left panel. **F** IDH1/2 and IDH3 activity in cell extracts obtained from HEK HyPer-DAO-NLS cells n.t. and after 50 mM D-ala for 20 min or 50 mM D-ala for 20 min and additional 24 hrs D-ala free incubation time. $n = 6$ independent experiments for n.t., L-ala and D-ala, $n = 5$ independent experiments for D-ala on-off. **G** Quantification of IDH3γ oxidation analyzed by PEG switch assay in HEK HyPer-DAO-NLS cells n.t. and after treatment with 50 mM D-ala ± 8 mM NAC for 20 min or 50 mM D-ala for 20 min and an additional 30 min D-ala free incubation time (on-off). $n = 5$ independent experiments. Representative experiment is shown in the left panel. Data are presented as mean values ± SEM, *$p < 0.05$, one-way ANOVA (C, E, F and G). Source data are provided as a Source Data file.

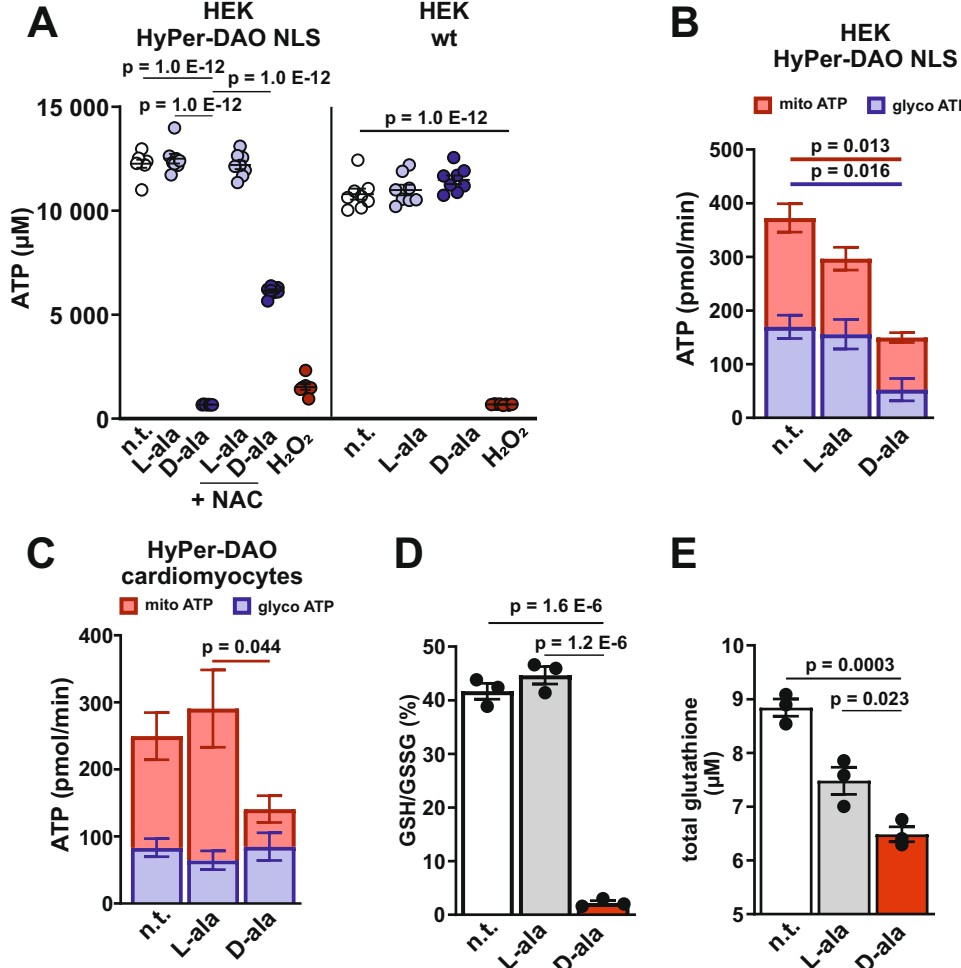

**Fig. 5 | Endogenously produced $H_2O_2$ impairs ATP produced in the mitochondria. A** ATP levels in HEK HyPer-DAO-NLS and wild type (wt) cells either non-treated (n.t.) or after treatment with 50 mM L-ala and D-ala ± 8 mM NAC or 500 µM $H_2O_2$ for 20 min, $n = 8$ independent samples. **B** Mitochondrial and glycolytic ATP production in HEK HyPer-DAO-NLS cells after treatment with 50 mM L-ala or D-ala for 20 min. $n = 4$ independent experiments. **C** Mitochondrial and glycolytic ATP production in

cardiomyocytes isolated from HyPer-DAO mice after treatment with 2 mM D-ala or L-ala for 20 min. $n = 7$ independent mice per group. **D** Ratio of reduced glutathione (GSH) to oxidized glutathione (GSSG) and total glutathione levels **E** in HEK HyPer-DAO-NLS cells either n.t. and after treatment with 50 mM D-ala or L-ala for 20 min, $n = 3$ biologically independent samples. Data are presented as mean values ± SEM, one-way ANOVA (A, B, C, D and E). Source data are provided as a Source Data file.

enzyme IDH3 as a specific $H_2O_2$ target protein in our study. In detail, as a major finding the γ-subunit of IDH3 was identified as a reversible redox switch indicating that $H_2O_2$-signaling and metabolism converge at the TCA cycle.

Since the $H_2O_2$ generating DAO enzyme was placed in the cell nucleus, we generated an intracellular $H_2O_2$ gradient. Detecting $H_2O_2$ levels in these cells by transient transfection of HyPerRed localized to the mitochondria indicated that the $H_2O_2$ levels in the

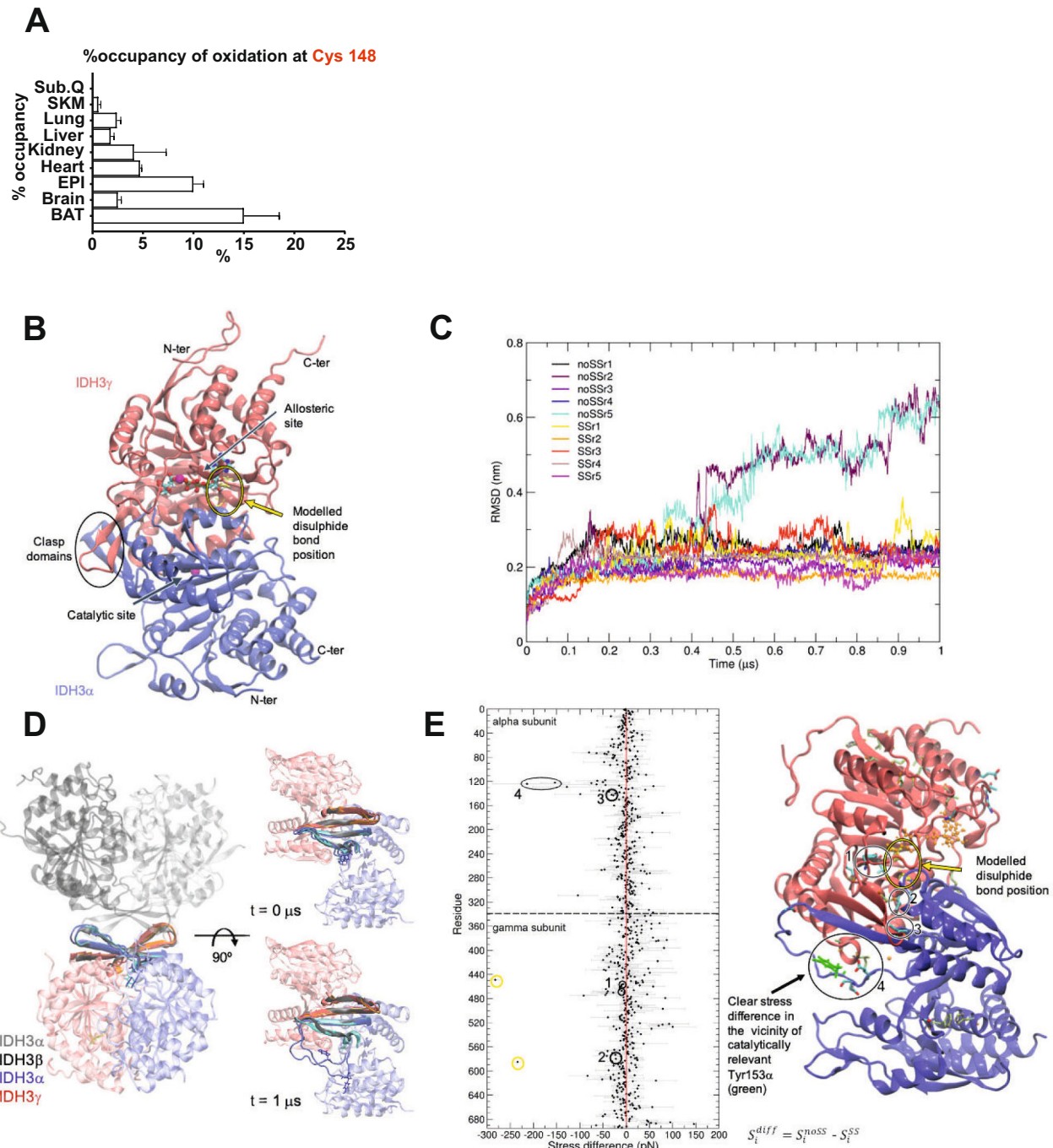

**Fig. 6 | Disulfide-bridge formation between IDH3γ Cys148 and Cys284 changes the tetramerization interface of IDH3. A** Quantification of IDH3γ Cys148 oxidation as the percentage of occupancy in different tissues from $n = 5$ biologically independent 16-week-old male C57BL/6 J mice. Data are obtained from the Oximouse database. subQ subcunateous fat, SKM skeletal muscle, epi epididymal fat, BAT brown adipose tissue. mean ± SEM. **B** Structure of the IDH3α/γ dimer with main functional sites. Citrate, Mg²⁺ and ADP are shown as ball and sticks. **C** Root mean square deviation (RMSD) plot for the clasp region for each system/replica. **D** Selected clasp regions from the two simulations showing substantial conformational changes (noSSr2, noSSr5, in blue for α subunit and red for γ subunit) and one stable reference (SSr2, in cyan for α subunit and orange for γ subunit)

superimposed onto the tetrameric structure of IDH3 (IDH3α/γ / IDH3α/β, PDB ID: 7CE3, clasp regions in gray). For clarity, the zoomed images do not show the IDH3α/β dimer. **E** Stress difference plot. Residues were renumbered consecutively from 0. The difference between the systems without (noSS) and with disulfide bridge (SS) was obtained by computing the difference between their averages, the standard deviations for each residue in the averaged curves noSS and SS were propagated to the difference (gray error bars). Residues with statistically relevant ($p < 0.0001$ in a Student's *t*-test on the two series with $n = 50$ data points) changes in the total stress between noSS and SS are highlighted and mapped onto the structure. Cysteines involved in the disulfide-bond are circles in yellow. This figure is related to Supplemental Fig. 4. Source data are provided as a Source Data file.

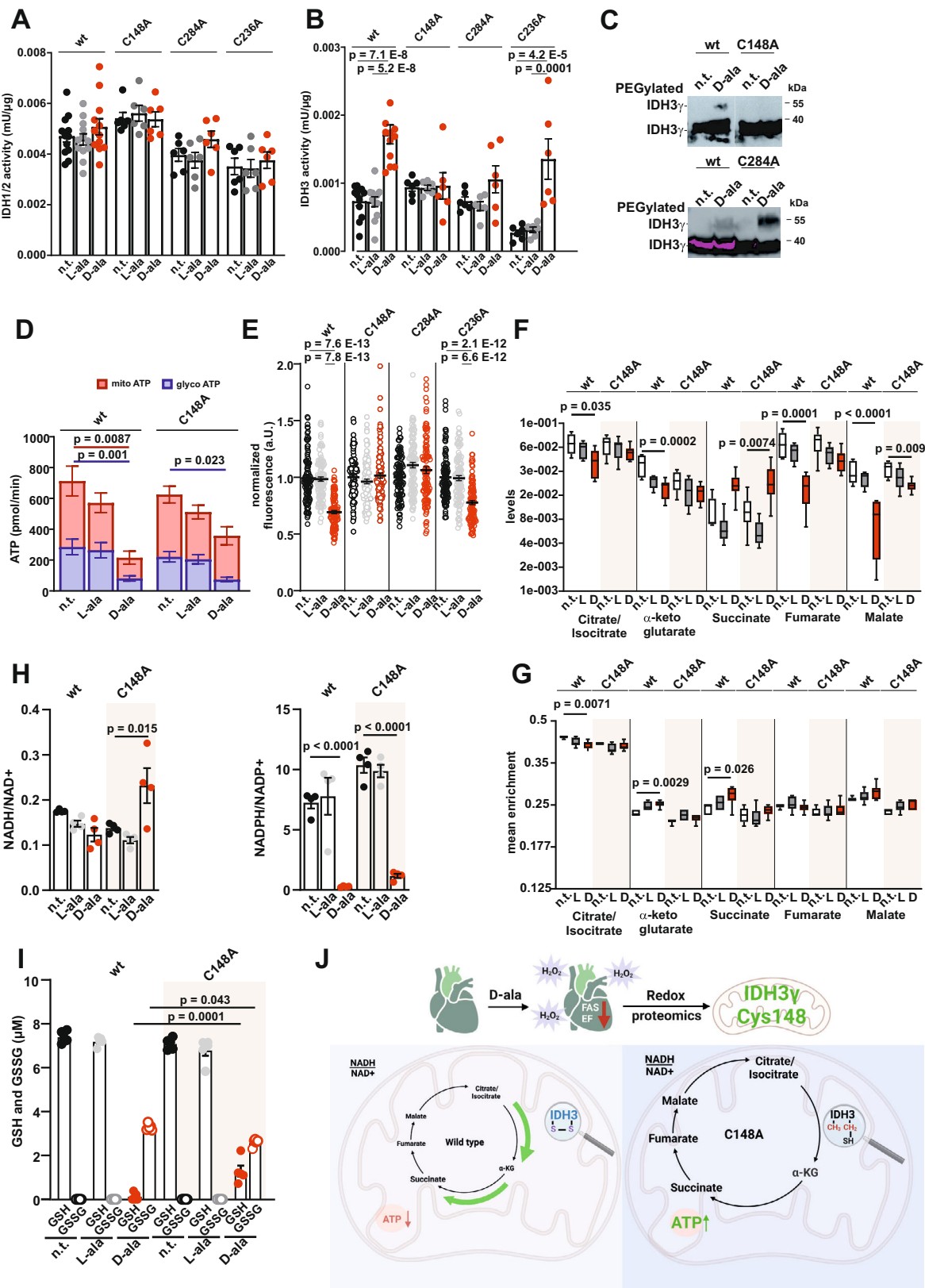

nucleus are indeed higher than in the mitochondria. Even more, when HyPer-DAO was localized to the mitochondria the same D-ala concentration resulted in less $H_2O_2$ levels compared to the nuclear localization, which indicates a high $H_2O_2$ buffer capacity of the mitochondria. This is in line with the previous characterization of the HyPer-DAO system[39,40]. The generation of transgenic mice

expressing the HyPer-DAO fusion protein in the cytoplasm or mitochondrial matrix was not successful (data not shown). Still we excluded in the HEK HyPer-DAO cells, in which we placed the fusion protein to the nucleus, cytoplasm or mitochondria that the localization matters for the modification of IDH3γ as well as the altered IDH3 activity and ATP levels.

**Fig. 7 | Redox modification of IDH3γ Cys148 and Cys284 is responsible for ATP production in the mitochondria.** IDH1/2 (**A**) and IDH3 (**B**) activity in cell extracts obtained from HEK HyPer-DAO-NLS wild type (wt), C148A, C284A and C236A cells non-treated (n.t.) and after treatment with 50 mM D-ala or 50 mM L-ala for 20 min, $n = 12$ independent experiments for HyPer-DAO-NLS wt cells and $n = 6$ independent experiments for C148A, C284A and C236A cells. **C** Representative immunoblot for IDH3γ oxidation analyzed by PEG switch assay in HEK HyPer-DAO-NLS wt, C148A and C284A cells n.t. or after treatment with 50 D-ala for 10 min. The experiment has been performed three times. **D** Mitochondrial and glycolytic ATP production in HEK HyPer-DAO-NLS wt and C148A cells after treatment with 50 mM D-ala or L-ala for 20 min. $n = 7$ independent experiments. **E** Mitochondrial ATP levels determined with the fluorescence sensor ATP-red in HEK HyPer-DAO-NLS wt, C148A, C284A and C236A cells n.t. and after treatment with 50 mM D-ala or L-ala for 20 min. $n = 100$ independent cells were analyzed per condition. **F, G** HEK HyPer-DAO-NLS wt and C148A cells were cultured with uniformly labeled $^{13}C$-glucose for 24 h. Subsequently cells were left n.t. or treated with 50 mM D-ala (D) or L-ala (L) for 20 min. Cells were lysed and total abundance of TCA cycle metabolite (**F**) and mean enrichment of $^{13}C$ labeled metabolites (**G**) were determined. $n = 5$ independent biological samples for HyPer-DAO-NLS cells n.t., $n = 6$ independent biological samples for C148A cells n.t. and $n = 8$ independent biological samples for HyPer-DAO-NLS and C148A cells either D-ala and L-ala treated. **H** NADH/NAD$^+$ and NADPH/NADP$^+$ ratios in HyPer-DAO-NLS wt and C148A cells n.t. or after treatment with 50 mM D-ala or L-ala. $n = 4$ independent biological samples per condition. **I** GSH and GSSG levels in HyPer-DAO-NLS wt and C148A cells n.t. or after treatment with 50 mM D-ala or L-ala. $n = 5$ independent biological samples per condition. **J** Summary scheme of the findings obtained in the HyPer-DAO mice and the HEK HyPer-DAO cells. Created with BioRender.com. Data are presented as mean values ± SEM (except **F, G**), Box-Whisker-Plot (showing the minimun, median and maximum with error bars showing the SD) in **F** and **G**, one-way ANOVA (**A, B, D, E, F, G, H** and **I**). This figure is related to Supplemental Fig. 5. Source data are provided as a Source Data file.

The modification of the IDH3γ pool as detected by PEG switch was not complete, which might be the basis for the reversibility upon release of the H$_2$O$_2$ challenge. The H$_2$O$_2$-altered IDH3 activity was further confirmed in independent cell lines including tumor cells and the myoblast C2C12 cells. The γ-subunit has regulatory function for the activity of the IDH3 tetramer and the redox modification identified in our mouse and cell models indeed results in altered enzyme function. There is some evidence from non-mammalians that IDH enzymes are redox modified[41–43]. In contrast to mammals, most bacteria and archaea however contain only NADPH-dependent homologs of IDH1/2. A redox-modification of IDH3 and its role in mammalian metabolism has not been described so far.

Mitochondria have their own specific thiol redox machineries[44], which overall could promote specific and reversible thiol modifications. Mitochondrial-derived ROS affect multiple cellular functions[45]. In addition, it is well-established that redox regulation of mitochondrial protein thiols is important for their function[46]. Within our redox-proteomic screen, we identified several mitochondrial protein candidates that demonstrated increased redox modification upon stimulation of endogenous cellular H$_2$O$_2$ production including TCA cycle enzymes. The TCA cycle is the principal means of aerobic energy metabolism from acetyl-CoA. A number of mammalian enzymes in the TCA cycle and the oxidation phosphorylation pathway have been shown to be redox-modified in the past. This includes the 2-oxoglutarate dehydrogenase[47], aconitase[48] and complex I, II and V[49,50]. SDH complex subunit A and B, which are one of four nuclear-encoded subunits that comprise complex II, were also present in the candidate list in our redox-proteomic screen. Analysis of SDH activity, however, revealed no change after D-ala treatment in our mouse or cell models. Redox modifications due to an increase in H$_2$O$_2$ levels are typically described to result in decreased activities of these enzymes. In consequence, this diminishes NADH production and therefore limits electron flow and thus further ROS release at the cost of ATP synthesis. This seems to be part of the pathophysiological events during ischemia-reperfusion injury[51]. We indeed saw evidence for a disturbed electron flow as indicated by elevated succinate levels. This is reminiscent of the extensive succinate accumulation during ischemia and its subsequent rapid oxidation on reperfusion, which drives ischemia/reperfusion injury[36,52]. The succinate accumulation as observed in our cells was independent of the identified IDH3γ modification since it was detected in the wt and in the C148A mutant cells. This indicates, as expected, that H$_2$O$_2$ most likely also affects other targets, which impact on metabolism aside from IDH3γ.

By profiling NF-κB and Nrf2 target genes in the hearts of the HyPer-DAO mice, we could show that the oxidative stress induced by the DAO enzymes was mild and reversible. In contrast to the mostly inhibiting effects by redox modification described so far, we found a gain of function in IDH3 in consequence of thiol modification. These functional analyses were paralleled by molecular simulations and analyses of cysteine point mutated cell lines. Altogether these data imply that IDH3γ Cys148 and Cys284 build a reversible disulfide-bridge that favors conformational changes in the catalytically active IDH3α-subunit. The altered IDH3 activity was found to be reversible upon removal of the H$_2$O$_2$ stimulus or by treatment with NAC. These effects were paralleled by a rescued function of the heart and ionotropic capacity of myocardial slices under the same conditions. Although we could also show IDH3γ Cys148 modification in hearts of TAC-treated mice, as a caveat it should be noted that a functional proof of the modification for disease progression in vivo is pending due to the lack of a respective cysteine edited mouse model.

Analysis in the IDH3γ C148A point mutated cells indicate that the disulfide-bridge and resulting increased IDH3 activity is consequently important for the H$_2$O$_2$-induced changes in mitochondrial ATP production. Our data indicate a link between redox signaling and the TCA cycle upon changes in intracellular H$_2$O$_2$ gradients. According to tracer studies performed, the flux from citrate/isocitrate to α-ketoglutarate and succinate was increased in line with the increased IDH3 activity in consequence of IDH3γ C148 modification. Levels of later intermediates (i.e., fumarate, and malate) downstream of IDH3 and ATP levels were reduced in consequence of H$_2$O$_2$ exposure. This was prevented in the C148A mutated cells and accompanied by a rebalancing of reducing NADH, NADPH an GSH equivalents. Thus, the on-off characteristic in IDH3 activity in response to H$_2$O$_2$ arms cells with a regulatory mechanism that allows metabolic adaptation to changes in the redox environment. Moreover, this might also indicate that at least some molecular changes that occur in H$_2$O$_2$ challenged cells can be reversed and cell and tissue integrity at least partly regained. This is of special importance in the context of heart failure.

The physiological importance of the reversible effect of H$_2$O$_2$ on IDH3γ was reflected by the fact that the D-ala-treated myocardium retained its inotropic capacity within two D-ala treatment-free days. The notion that endogenous H$_2$O$_2$ production did not produce a permanent phenotype was further underscored by the lack of cardiac fibrosis in D-ala-treated mice. This characteristic is similar to post-ischemic myocardial dysfunction also named stunning[53]. H$_2$O$_2$ was thought for long to play a major causative role in the stunned myocardium[54], however the exact mechanism remains speculative and still represents one of the major unresolved issues pertaining to the pathogenesis of myocardial stunning. Using the HyPer-DAO mice, we prove that in conscious mice stimulation of endogenous production of H$_2$O$_2$ leads to a stunning-like phenotype associated with specific changes in cellular metabolism. We further underscore the physiological importance of the identified IDH3γ modification by showing that this modification can be found in left ventricular samples from TAC-challenged mice and as a major finding deliver IDH3γ as a direct molecular redox switch.

# Methods

## Ethics statement

The study complied with all the ethical regulations for work with experimental animals. Animal experiments were carried out according to the guidelines for the care and use of laboratory animals, following directive 2010/63/EU of the European Parliament. The animal work with the HyPer-DAO transgenic mice and the animal protocol regarding transverse aortic constriction were confirmed by the institutional guidelines and were approved by the Niedersächsische Landesamt für Verbraucherschutz und Lebensmittelsicherheit, Oldenburg Germany (33.9-42502-04-16/2352 and 33.9-42502-04-16/2102).

## Generation of HyPer-DAO transgenic mice

The HyPer-DAO fusion used for generating transgenic mice was the same as used before and contained the Gly-Gly-Ser-Gly linker between HyPer and DAO[15]. The mαMHC-HyPer-DAO-NLS-hGHpolyA plasmid was linearized with the restriction enzyme EcoRV (Thermo Fisher). The linearized plasmid was purified with the Monarch DNA purification kit (New England Biolabs). Transgenic mice were generated by pronuclear blastocyst injection in C57Bl6N mice (Jackson Laboratories). The injection was performed by the Core Facility of the Max Planck Institute for Experimental Medicine, Göttingen. Both adult male and female mice (at the age of 8–14 weeks) were used for the experiments. Mice were genotyped by a standard polymerase chain reaction using the following primers: 5′-CCTGAGACCCGCCAGGAGAG-3′ and 5′-GGATCGCTCTCCCTAGCTGC-3′, resulting in a 254-bp fragment. Transgenic mice were mated with wt mice to produce heterozygous offspring and wt littermate controls.

## D/L-ala treatment of mice

D-alanine (D-ala) and L-alanine (L-ala) were purchased from Sigma-Aldrich. 55 mM D-ala or L-ala were used in the drinking water for the mice in the in vivo experiments. The drinking water was exchanged every second day.

## Echocardiography

Left ventricular dimension in systole and in diastole (left ventricular inner diameter in systole and diastole, LVIDs/LVIDd and enddiastolic volume, EDV), fractional area shortening, anterior and posterior wall thickness, and ejection fraction were measured by transthoracic echocardiography as described[55,56]. In brief, mice were anesthetized using an induction chamber with a flow of 3% isoflurane (Forene, Abbvie) in presence of oxygen, followed by 1% isoflurane maintained with a face mask on the measurement table. Two-dimensional image tracings were recorded from the parasternal long- and short-axis view at midpapillary level, M-mode was acquired in the short-axis view (Vevo 2100 system, Visual Sonics Inc, Toronto, Canada).

## Transverse aortic constriction

Aortic pressure overload was induced by a minimally invasive transverse aortic constriction (TAC) procedure on 10 weeks old C57BL/6 N mice as described previously[57]. Mice were anesthetized by intraperitoneal injection of medetomidin (0.5 mg/kg body weight), midazolam (5 mg/kg body weight), and fentanyl (0.05 mg/kg body weight) prior to opening the chest through a small suprasternal surgical approach. Constriction of the transverse aorta was induced using a 27 G spacer and a 5.0 polyviolene suture. Sham operated animals underwent the same surgical procedure but without aortic constriction. After suturing the wound, anesthesia was antagonized by subcutaneous injection of atipamezol (2.5 mg/kg BW) and flumazenil (0.5 mg/kg BW). Post-operative analgesia contained the subcutaneous injection of buprenorphin (0.05–0.1 mg/kg BW) and carprofen (5 mg/kg BW). The aortic pressure gradient was measured 2 days post TAC by transthoracic echocardiography using a 20 MHz probe (Vevo 2100, VisualSonics). TAC mice included in this study showed an aortic pressure gradient

>40 mmHg. Animals were sacrificed 4 weeks after TAC, hearts were isolated and stored in liquid nitrogen for further analysis.

## Force measurement

Mice were killed by cervical dislocation and hearts were explanted, retrogradely flashed with ice-cold PBS and the ventricles placed in ice-cold slicing buffer (140 mM NaCl, 6 mM KCl, 1 mM MgCl₂, 1 mM glucose, 10 mM Hepes, 5 mM BDM, pH 7.4). The left ventricle was cut along the middle of the septum and embedded in 4% low-melting agarose (Sigma-Aldrich, USA). 300 μm thick slices were cut. Out of these slices 0.5 cm² tissue squares were prepared, glued (histoacryl glue, B. Braun, Germany) on two plastic triangles and placed into vertical 20 mL glass organ baths filled with 37 °C Krebs solution bubbled with carbogen (95% $O_2$ and 5% $CO_2$). Plastic triangles were fixed to the hooks of FT20 transducers and core holders with field stimulation electrodes (Hugo Sachs Elektronik, Germany) on each site and pre-stretched to 2–3 mN. Isometric force was measured with a KWS 3073 bridge amplifier (HBM, Germany) and recorded with a Powerlab 8/35 using the LabChart7.2 software (ADInstruments). Electrical pacing was performed with a Myotronic stimulator (Myotronic) at 0.5 Hz until diastolic force and force generation stabilized. Finally, the pre-stretch was readjusted to 2–3 mN, the pacing frequency was increased to 2 Hz and the experiments started after an additional 15–20 min. Indicated slices were pre-incubated with 1 mM NAC for 15 min and D-ala (0.1 mM, 1 mM, 10 mM) was added in a 15 min long step. To quantify isometric force, 10–15 peaks without arrhythmic events at the end of each 15 min step were averaged. These values were normalized to the highest force generated by each slice.

## Sirius red/fast green

Sections were incubated with a dye combination of Sirius Red and Fast Green (Chondrex) for 30 min. Sections were washed with ddH₂O and dehydrated with ethanol (EtOH 60% 2 min, EtOH 75% 2 min, EtOH 96% 2 min, EtOH 99% 2 min). After mounting (Rotihisto kit II, Roth), sections were analyzed using the Observer D1 microscope (Zeiss). Image J-win64 (v1.48) was used for alignment of the multi-tile images (linear blending, regression threshold: 0.30, overlap settings: 20%). The percentages of total, perivascular and interstitial fibrosis were compared to the total amount of tissue within an image by determining the total tissue areas occupied by cardiomyocytes and collagen but excluding the lumen (i.e., empty spaces). The original images were converted to an RGB stack. The area occupied by tissue was selected and the threshold was applied to delineate the fibrotic area over the entire area.

## Proteomics and redox-proteomics

Tissue homogenization and protein extraction: heart tissue (10 mg) was homogenized using a bead mill homogenizer in 500 μL of modified RIPA buffer (mRIPA; 50 mM Tris HCl, 150 mM NaCl, 1 mM EDTA, 1% NP-40, 0.25% sodium deoxycholate, 1 mM NaF) supplemented with 10 mM MSTP ((4-(5-methanesulfonyl-[1,2,3,4]tetrazol-1-yl)-phenol); Xoder Technologies) to selectively block reduced thiols[58]. After 30 min incubation on ice, the tubes were centrifuged at 18,000 g and 4 °C for 10 min, the supernatant was transferred to a new tube and the protein concentration was measured using the bicinchoninic acid (BCA) assay. 100 μg protein fraction was precipitated with 4x cold acetone (volume) overnight at −20 °C.

Reduction of reversible oxidized thiols: the protein pellet was isolated by centrifugation at 16,000 g at 4 °C for 5 min, resuspended in 150 μL buffer (50 mM HEPES,1 mM EDTA, 0.1% SDS, pH 7.5), and reduced using 10 mM DTT (final concentration). Following incubation at 37 °C with shaking (850 rpm) for 1 h, the excess DTT was removed by passing the protein extract through a Bio-Gel P6 spin column.

Enrichment of thiol-containing proteins: proteins containing newly released thiols were isolated by covalent binding (4 °C with up

and down rotation overnight) to thiopropyl sepharose resin pre-conditioned with the binding buffer (50 mM HEPES and 1 mM EDTA, pH 7.5). To remove the non-covalently bound proteins, the resin was washed five times with each of the following solutions: 8 M urea, 2 M NaCl, 0.1% SDS in PBS, 80% (vol/vol) ACN and 0.1% (vol/vol) TFA, and 50 mM $NH_4HCO_3$.

On-resin tryptic digestion and elution of enriched cysteine-containing peptides: proteolytic digestion was performed on the resin by adding 200 μL 50 mM $NH_4HCO_3$ and trypsin (trypsin: protein = 1:20; Pierce Trypsin Protease, MS Grade, Thermo Scientific) and overnight incubation at 37 °C. After digestion, the resin was similarly washed with 8 M urea, 2 M NaCl, 80% ACN and 0.1% TFA, and 50 mM $NH_4HCO_3$, and the covalently bound peptides were eluted by incubating three times with 20 mM DTT in 25 mM $NH_4HCO_3$ and shaking at 850 rpm for 30 min at room temperature. Each eluent was collected by spinning at 1.500 g for 1 min at room temperature. Finally, the resin was washed with 100 μL of 80% ACN/0.1% TFA to collect the residual peptides. The eluent fractions were combined, dried using SpeedVac, and stored at −80 °C until LC-MS/MS analysis.

LC-MS/MS analysis and database search: dried peptides were dissolved in water containing 0.1% formic acid and 5% ACN. Samples were analyzed on an Orbitrap Eclipse Mass Spectrometer coupled with a Vanquish Neo nano-UHPLC system (Thermo Scientific, Waltham, MA, USA) via the FAIMS (high-field asymmetric waveform ion mobility spectrometry) Pro interface. Peptides were separated on a DNV Pep-Map Neo (1500 bar, 75 μm x 500 mm, Thermo Scientific, Waltham, MA, USA) column for 120 min employing linear gradient elution consisted of water (A) and 80% acetonitrile (B) both of which contained 0.1% formic acid. Data were acquired using top-speed data-dependent scan where maximum number of MS2 spectra were collected from fragmentation of selected precursor ions per 3 s of cycle time between adjacent survey MS1 spectra. MS2 scan was repeated with precursor ion subsets isolated by FAIMS at three different compensation voltages of −45 eV, −55 eV, and −65 eV sequentially. Dynamic exclusion option was enabled which duration was set to 120 s. To identify proteins, spectra were searched against the UniProt mouse protein FASTA database (17,082 annotated entries, Oct 2021) using the Sequest HT search engine with the Proteome Discoverer v2.5 (Thermo Scientific, Waltham, MA, USA). Search parameters were as follow: parent mass error tolerance of 10 ppm; fragment mass error tolerance of 0.6 Da (monoisotopic); maximum missed cleavage with trypsin at 2 sites; variable modifications of +15.995 Da (oxidation) on methionine, and +160.039 Da (MSTP) on cysteine. The identified peptides were validated using a reversed database search in percolator node under the parameters of 0.05 for maximum delta Cn, 0.01 and 0.05 for each strict and relaxed target FDR.

Samples were compared by fold change of cysteine-containing peptide abundance (i.e. peak intensity) calculated by label-free quantitation (LFQ). Peptide abundance was normalized to the total ion current (TIC), and all the missing values was replaced with one fifth of minimum abundance in each peptide feature. Peptides with greater than 50% of missing values were excluded from comparison.

## ATP/ADP in tissue samples
Levels of adenosine diphosphate (ADP), adenosine triphosphate (ATP) in tissue samples were determined by Anion-Exchange Chromatography coupled to Electrospray Ionization High-Resolution Mass Spectrometry (IC-ESI-HRMS) using a procedure previously described[59] with several modifications: Frozen mouse heart tissue samples were suspended in ice-cold acetonitrile/methanol/water 2:2:1 (v/v/v) using the Precellys 24 Homogenisator (Peqlab, Erlangen, Germany) at 6500 rpm for 10 s. The protein content of the homogenate was routinely determined using bicinchoninic acid. 100 μL of homogenate were mixed with further 225 μL of acetonitrile/methanol/water 2:2:1 (v/v/v) and 25 μL of a mixture of isotope-labeled internal standards in Milli-Q

water (50 μM 13C10-ATP, Sigma-Aldrich). After thorough mixing and centrifugation (16,100 RCF, 5 min, 4 °C), 300 μL of supernatant were dried under reduced pressure. The residue was resolved in 100 μL of Milli-Q water, transferred to autoinjector vials and immediately measured. IC-HRMS analysis was performed using a Dionex Integrion RFIC system (Thermo Scientific) equipped with a Dionex IonPac AS11-HC column (2 mm × 250 mm, 4 μm particle size, Thermo Scientific) and a Dionex IonPac AG11-HC guard column (2 mm × 50 mm, 4 μm, Thermo Scientific) and coupled to a Q Exactive HF quadrupole-orbitrap mass spectrometer (Thermo Scientific). 5 μL of sample were injected using a Dionex AS-AP at 10 °C. The IC was operated at a flow rate of 0.38 mL/min with a potassium hydroxide gradient which was produced by an eluent generator with a potassium hydroxide cartridge and Milli-Q water. The gradient started with 10 mM KOH over 3 min, 10 – 50 mM from 3 to 12 min, 50 – 100 mM from 12 to 19 min, held at 100 mM from 19 to 25 min, and re-equilibrated at 10 mM for 3 min. The total run time was 28 min. An Dionex ADRS 600, 2 mm suppressor was operated with 95 mA, and methanol was used to produce a make-up flow at a flow rate of 0.15 mL/min. The mass spectrometer was operated in the negative ion mode. Full MS scans in the range of m/z 60-900 were acquired with a resolution of 120,000, an Automatic Gain Control (AGC) target value of $1 × 10^6$ and a maximum injection time (IT) of 240 ms. Spectrum data were collected in the centroid mode. The ESI source was operated with flow rates for sheath gas, auxiliary gas, and sweep gas of 50, 14 and 3, respectively. The spray voltage setting was 2.75 kV, the capillary temperature 230 °C, the S-lens RF level 45, and the auxiliary gas heater temperature 380 °C[59]. The exact m/z traces of the internal standards and endogenous ADP and ATP were extracted and integrated using Skyline 21.2.0.369 (open-source). Endogenous metabolites were quantified by normalizing their peak areas to those of the internal standards: 13C10-ATP was used as internal standard for endogenous ADP and ATP. Endogenous ADP and ATP were quantified by normalizing their peak areas to the peak area of the internal standard 13C10-ATP. This peak area ratio was normalized to the protein content of the homogenate.

## Isolation of cardiomyocytes
Adult ventricular cardiac myocytes were isolated via the Langendorff perfusion as described previously[55].

## Cell culture
HEK293 (ATCC, CRL-1573), HEK293T (ATCC, CRL-3216), MDA-MB-231 (ACC 732) and Hep3B (HB-8064) cells were obtained from Deutsche Sammlung von Mikroorganismen und Zellkulturen (DSMZ) and American Type Culture Collection. C2C12 myoblasts were purchased from Sigma Aldrich (ECACC 91031101). Cells were cultivated in DMEM, 10% fetal calf serum.

## Generation of stably transduced cell lines
HEK and Hep3B cells expressing the Hyper-DAO fusion protein in the nucleus (NLS), the cytoplasm (NES) and in the mitochondria (MLS) were generated by lentiviral transduction using pCMVpL4-HyPer-DAO-3xNLS, pCMV-pL4-HyPer-DAO-1xNES and pCMV-pL4-HyPer-DAO-Cox8MLS, respectively. Viral particles were produced in HEK293T cells using the ViraPower lentiviral packaging mix according to the manufacturer's instructions (Thermo-Fischer Scientific). Puromycin (Sigma-Aldrich) was used to select HEK and Hep3B cells which were successfully integrated with the plasmid. A monoclonal cell population was selected for each cell line by limiting dilution.

## Gene editing
Single guided RNAs (sgRNAs) were designed using the online tool CRISPOR (v5.01). The homology-directed repair (HDR) templates for each point mutation were designed as single-stranded oligodeoxynucleotides (ssODNs) along with silent mutations at the PAM sites.

sgRNAs and HDR templates were ordered as Alt-R™ Oligos from Integrated DNA Technologies. Cas9 nuclease, HDR enhancer and electroporation enhancer were purchased through Integrated DNA Technologies. Following sequences were used: sgRNA, Cys148: 5′-CG CCTGGAAGGCTCTTACAG-3′, sgRNA Cys 284 5′-CATCGTCAACAATG TCTGCG-3′, sgRNA Cys236: 5′- CCACCTCCCTGCAGCACTGG-3′; HDR template Cys148: 5′-CTCCCGGACAATGAGGGATGTCTATGTCCTTGTG CCGGGTCACCACGCCTGGAAGGCTCTTGGCGTGAATGACGTTGGCAT AGAGGTCCAGGCTGGTGCTGGGAGGGGACGGAGAAAGAGGCTG-3′,H DR template Cys284: 5′-CCCCAGCAGTTTGATGTCATGGTGATGCC-CAATCTCTATGGCAACATCGTCAACAATGTCGCCGCTGGACTGGTCG GGGGCCCAGGCCTTGTGGCTGGGGCCAACTATGGCCATGTGTACGC G-3′, HDR template Cys236: 5′- CTGGTGTCCTATTCCTAACCCCCCAC-CAGGAAACTGGGCGATGGGCTTTTTCTCCAGTGCGCCAGGGAGGTGG CAGCCCGCTACCCTCAGATCACCTTCGAGAACATGATTGTGGATAAC ACC-3′, HDR template Cys148/Cys284: 5′-GCGGCACTGCCACCGG-CACTCACTGTTTCAAACACCGCGTACACATGGCCATAGTTGGCCCCA GCCACAAGGCCTGGGCCCCCGACCAGTCCAGCGGCGACATTGTTGA CGATGTTGCCATAGAGATTGGGCAT-3′. Each sgRNA was prepared by duplexing Alt-R CRISPR-Cas9 crRNA and Alt-R CRISPR-Cas9 tracrRNA. The ribonucleoprotein (RNP) was made by adding Alt-R S.p. Cas9 Nuclease V3 (Integrated DNA Technologies). Cells were electroporated with 5 µL RNP complex, 1.2 µL HDR donor oligo, 1.2 µL Alt-R Cas9 Electroporation Enhancer, and 2.6 µL PBS using the 4D-Nucleofector module of 4D-Nucleofector (Lonza, AAF-1003X) according to the manufacturer's specifications. Single cell colonies were selected and expanded. Genomic DNA was isolated and the successful gene editing in IDH3γ was confirmed by Sanger sequencing.

## HyPer measurements
For HyPerGreen measurements cells were excited at 420 and 500 nm and the emitted light was recorded at 510 nm using the IX83 microscope and the MT20 illumination system (Olympus). Light intensity was set to 11% for isolated cardiomyocytes, 4% for HEK cells and the exposure time was set to 1 s. Each image was recorded at an interval of 120 s. After reaching stable baseline readings, D-ala, L-ala, $H_2O_2$ (Carl Roth)‚ menandione, rotenone or antimycin A (Sigma-Aldrich) were added. In some experiments stably transduced HEK cells were transiently transfected with pC1-HyPerRed-mito plasmid[60] by lipofection, cells were excited at 550 nm and the emitted light was recorded at 590 nm.

## MitoSOX measurements
Cells were loaded with 5 µM MitoSOX™ red (Invitrogen, M36008) at 37 °C for 10 min, protected from light. After the cells were washed twice with 137 mM NaCl, 2.7 mM KCl, 1.8 mM $CaCl_2$, 1 mM $MgCl_2$, 0.2 $Na_2HPO_4$, 12 mM $NaHCO_3$, 5.5 mM glucose, imaging was carried out using an inverted fluorescence microscope IX83 and the CellSens software (Olympus, v1.18). MitoSox was excited at 550 nm using the MT20 illumination system (Olympus) with an exposure time of 1 s. The emitted light was detected via a CCD camera at 590 nm. Images were taken every 2 min for 20 min. The baseline of the fluorescence was measured in the first three images.

## Mitotracker-live cell staining
Cells were incubated with Mitotracker (Orange CMTM Ros, M7510, Thermo Fisher, dilution 1:8000) for 15 min. After being washed with PBS, the cells were incubated with Draq5 (DR50050, Biostatus, dilution 1:1000) for 10 min. Cells were analyzed by confocal microscopy using the LSM800 (Zeiss).

## Western blots
Tissues were lysed in 3.7 M urea, 135 mM TRIS pH 6.8, 1% SDS, 2% NP-40, protease inhibitor cocktail (Roche). Protein extracts were separated by SDS gel electrophoresis and transferred to nitrocellulose membranes. Western blots were performed using the following primary and secondary antibodies: anti-GFP (1:1000, 11814460001, Sigma-Aldrich), anti-GAPDH (1:1000, 2118 L, Cell Signaling), anti-IDH3γ (1:1000, HPA002017, Sigma-Aldrich), anti-vinculin (1:10000, V9264, Sigma-Aldrich), goat anti-rabbit HRP (1:10000, sc-2004, Santa Cruz) and goat anti-mouse HRP (1:1000, 12-349, Sigma-Aldrich).

## PEG switch assay
The PEG switch assay was performed according to a previously described protocol[61]. In brief, cells were treated with 50 mM D-ala, 50 mM L-ala, 8 mM NAC or 500 µM $H_2O_2$ respectively. After a washing step with ice-cold PBS containing 20 mM of N-ethylmaleimide (NEM), 150 µL of ice-cold PEG switch lysis buffer containing 1% SDS, 100 mM NEM and 100 mM Tris pH 7.4 were added to the cells. Each lysate was incubated in a shaker at 50 °C and 600 rpm for 25 min to block free thiol groups by alkylation with NEM enriched lysis buffer. While 50 µL from each sample were used as inputs, the rest 100 µL were supplemented with buffer containing 200 mM DTT to reduce oxidized protein thiols. After a 30 min incubation at room temperature, excess DTT was removed using Zeba™ Spin desalting columns (Thermo-Fisher Scientific). After addition of the thiol-labeling buffer containing 70 mM PEG-maleimide (5 kDa; number 63187; Sigma-Aldrich) and 7% SDS in 500 mM Tris pH 7.4, samples were incubated on a laboratory agitator at room temperature for 2 hrs in the dark. Finally, each input and sample was supplemented with 6× reducing Laemmli sample buffer containing 5% ß-mercaptoethanol and subjected to Western immunoblot analysis.

## RNA quantification
Total RNA was isolated from left ventricles using Trizol (Invitrogen)/ chloroform extraction. 1 µg RNA was reverse transcribed into cDNA using the First Strand cDNA Synthesis kit (Thermo-Fischer Scientific) according to the manufacturer's instruction. Quantitative real-time polymerase chain reaction (qRT-PCR) was performed amplifying 1 µL cDNA with SensiMix TM SYBR Lox (Bioline), and 20 pM respective forward and reverse primers in an MX3005Pro light cycler (Agilent). To calculate the relative expression of fold difference in the gene of interest, the $\Delta\Delta CT$ analysis method was used. The murine ribosomal protein S12 (ms12) was used as a reference gene.

The following primers were used **ICAM1** for 5′-CCGTGGGGAG-GAGATACTGAGC-3′, rev 5′-GGAAGATCGAAAGTCCGGAGGC-3′; **IL-1β** for 5′-AGCTTCAGGCAGGCAGTATCAC-3′, rev 5′-AATGGGAACGTCA-CACACCAGC-3′; **TNFα** 5′-GACCCTCACACTCAGATCATCTTC-3′, 5′-CC ACTTGGTGGTTTGCTACGA-3′; **iNOs** for 5′-AAGTCCAGCCGCACCAC CCT-3′, 5′-TCCGTGGCAAAGCGAGCCAG-3;′ **NQO1** for 5′-GGTAGCGGC TCCATGTACTCTC-3′, rev 5′-CCAGACGGTTTCCAGACGTTTC-3′; **HMOX1** for 5′-GACAGCCCCACCAAGTTCAAAC-3′, rev 5′-CCTCTGA CGAAGTGACGCCATC-3′; **Srxn1** for 5′-CGGTGCACAACGTACCAATCG C-3′, rev 5′-GCAGCCCCCAAAGGAATAGTAG-3′; **Txnrd1** for 5′-GTGGG TTGCATACCTAAGAA-3′, rev 5′-GCATTCTCATAGACGACCTT-3′; **Prdx1** 5′-GGCTCGACCCTGCTGATAG-3′, rev 5′-GCAATGATCTCCGTGGGAC A-3′; **Prdx2** for 5′-GGGCATTGCTTACAGGGGTC-3′, rev 5′-TGGGCTTGA TGGTGTCACTG-3′; **Prdx3** for 5′-GAAGGTTGCTCTGGTCCTCG-3′, rev 5′-GGTGTGGAAAGAGGAACTGGT-3′; **Prdx5** for 5′-CCTTTGGGAAGGC-GACAGA-3′, rev 5′-CATCTGGCTCCACGTTCAGT-3′; **Gpx1** for 5′-CAGTC CACCGTGTATGCCTTC-3′, rev 5′-TCATTCTTGCCATTCTCCTGGT-3′; **Gpx2** for 5′-CCAGTTCGGACATCAGGAGAA-3′, rev 5′-GTCATGAGG GAGAACGGGTC-3′; **Gpx3** for 5′-AGCCAGCTACTGAGGTCTGA-3′, rev 5′-GAGGGCAGGAGTTCTTCAGG-3′; **Txn1** for 5′-GCCCTTC TTCCATCTCTCCTG-3′, Txn1 rev 5′-AGGTCGGCATGCATTTGACT-3′; **Txn2** for 5′-TTCCGGACTTTCATGCACAGTGGT-3′, rev 5′-TCCCCTCC ACAAACTTGTCCACC-3′; **IDH3α** for 5′-AGGGAAGTTGCGGAGAACTG -3′, rev 5′-GGGCTGTTCCATGAACCGAT-3′; **IDH3β** for 5′-GGAAGTGTT CAAGGCTGCTG-3′, rev 5′-ACGCCTCAGCTGCATATCAT-3′; **IDH3γ** for 5′-GCTGCAAAGGCAATGCTCAA-3′, rev 5′-GGAGGAATTGTTTGTTGT

GAGGA-3′; **mS12** for 5′-GAAGCTGCCAAGGCCTTAGA-3′, rev 5′-AA CTGCAACCAACCACCTTC-3′.

### $^{13}C_6$-glucose tracer analysis

After 24 hrs of seeding, the cells were incubated in DMEM containing uniformly labeled $^{13}C$ glucose for next 24 hrs. Subsequently, the cells were treated with 50 mM L-ala or D-ala respectively for 20 min. At the end of the treatment period, the cell culture supernatant was removed, cells were rapidly washed three times with ice-cold PBS and immediately harvested by scraping with 1 mL of ice-cold 80% methanol. The cell suspension was transferred to an Eppendorf cup and the cell culture dish was washed once with 500 μL of 80% methanol to make sure that the complete sample is collected. The wash was combined with the initial extract and the samples were stored at −80 °C until further analysis. For processing, samples were thawed and centrifuged at 10.000 g and 4 °C for 5 min. The supernatant was collected, and the protein pellet was washed twice with 200 μL of 80% methanol using 15.000 g for the last centrifugation step. The washes were combined with the extract and the sample was evaporated to dryness using an infrared vortex vacuum evaporator (CombiDancer, Hettich AG, Baech, Switzerland). The dried sample was reconstituted in 100 μL $H_2O$, 50 μL were transferred to a flat-bottom insert in a 1.5-mL glass vial and again dried for subsequent GC-MS analysis. Intermediates of glycolysis and TCA cycle were analyzed by GC-MS after derivatization using methoximation and silylation[62] employing an Agilent model 6890 GC (Agilent, Palo Alto, USA) with a MSD model 5975 Inert XL and a MPS-2 Prepstation sample robot (Gerstel, Muehlheim, Germany) for automated derivatization. As a quality control, 10 μL of an undecanoic acid (C11) solution (1 mM in isooctane) was added to each sample prior to silylation. An RXI-5MS column (30 m × 0.25 mm ID × 0.25 μm film thickness; Restek, Bad Homburg, Germany) equipped a 2 m precolumn was used. The initial column temperature was set at 50 °C (1 min), ramped at 5 K/min to 120 °C, then, at 8 K /min to 300 °C and held for 5 min. An injection volume of 1 μL and splitless injection was employed. Peak areas of all isotopologues were determined using MassHunter−Quantitative Analysis (version B.07.01, Agilent). The raw data were corrected for natural stable isotope abundance and tracer impurity using IsoCorrectoR (v0.1.13). Data are given as mean isotopic enrichment or as normalized peak areas to evaluate metabolite abundance. For the latter, peak areas of all isotopologues belonging to a metabolite were summed after correction for natural isotope abundance and tracer impurity. The summed peak area was then normalized to peak area of the derivatization standard C11 and the protein amount of the sample. To determine the protein amount, the protein pellet remaining after extraction was dissolved in $NaH_2PO_4$ buffer (20 mM with 1.2% sodium dodecyl sulfate) and subjected to protein quantification using the fluorescent dye SERVAPurple (Serva, Heidelberg, Germany).

### Glutathione levels

Glutathione levels were determined using the GSH/GSSG-Glo™ assay kit (V6611, Promega) according to the manufacturer's instructions.

### $NADH/NAD^+$ and $NADPH/NADP^+$ levels

$NADH/NAD^+$ and $NADPH/NADP^+$ levels were determined using the Glo™ assay kits (G9071 and G9081, Promega) according to the manufacturer's instructions.

### ATP level measurement by luminescence

ATP levels were determined using the CellTiter-Glo kit (G7570, Promega) according to the manufacturer's instructions.

### ATP level measurement by staining and fluorescence microscopy

Cells were washed twice with fresh medium and loaded with 5 μM ATP-Red dye (Merck, SCT045) for 20 min at 37 °C. After washing, the cells were excited at 550 nm using an inverted fluorescence microscope IX83, MT20 illumination system (Olympus) with an exposure time of 1 s. The emitted light was detected via a CCD camera at 570 nm. Fluorescence intensities of the intracellular ATP were calculated by the CellSens software. Generally, 25 regions of interest (ROI) for ATP labeling were selected on three consecutive images. Background was subtracted and the average pixel intensity of each ROI was calculated and extracted from the software. The average fluorescence intensity of each ROI was calculated as the mean value of the ROIs at the three time points. To normalize the data from different experiments, the data of the cells either treated with L-ala or D-ala (F(l), F(d)) was normalized by the data of the cells without any treatment(F(n)) as follows: Ratio = F(l)/F(n) or F(d)/F(n).

### ATP level measurement by oxygen consumption rate (OCR) and extracellular acidification rate (ECR)

Real-time measurements of OCR and ECR were carried out using the XFe-96 Extracellular Flux Analyzer (Seahorse, Agilent). Cells were plated in XFe-96 plates (Agilent) at the density of 30.000 cells/well. Cells were analyzed 24 hrs after plating. OCR and ECR were measured in XFe media (non-buffered Dulbecco's modified eagle medium (DMEM)) medium, pH-7.4 containing 10 mM XF glucose, 2 mM XF glutamine, 1 mM XF pyruvate under basal conditions and in response to 3.7 μM oligomycin and 1.3 μM of antimycin-A and rotenone (Sigma-Aldrich, St. Louis, MO, USA). OCR, ECR and ATP production from glycolysis and mitochondria were calculated according to the manufacturer's instructions.

### Enzyme activity measurements

For determining enzyme activities, the IDH1/2 and 3 (MAK062), succinate dehydrogenase (MAK197), malate dehydrogenase (MAK196), fumarase (MAK206) activity assay kits were used according to the manufacturer's recommendation (all Sigma-Aldrich).

For analysing IDH1/2 and IDH3 activities, at the end of the treatment, cells were lysed with the assay buffer provided in the kit. The lysates were then centrifuged at 13000 g for 10 min at 4 °C to remove cellular debris. Protein concentrations of the samples were estimated using the DC protein assay kit (Bio-Rad). Different protein amounts were tested to understand the kinetics of the enzymatic reaction. Isocitrate which is the common substrate for both IDH1/2 as well as IDH3 isoforms, along with $NADP^+$ as cofactor for IDH1/2 and $NAD^+$ as cofactor for IDH3, respectively were added. This reaction mixture was added 1:1 to the sample, followed by measuring the absorbance at 450 nm over a time duration of 55 min. The IDH1/2 and 3 activities showed a linear relationship to the protein input using protein concentrations ranging from 10-100 μg suggesting that the substrate concentrations used are not limiting the enzymatic activity. To exclude the activity resulting from endogenous substrates in the lysates, a blank reaction was carried out which contains the sample along with the reaction mixture as mentioned above, however without isocitrate, $NAD^+$ or $NADP^+$ added. The enzymatic activity of blank reaction was subtracted from the respective sample's activity. Standard curves were used to ultimately calculate the amount of the NADH or NADPH present in the sample and thus its enzyme activity. For calculating the enzyme activity per μg of the protein used, we adapted the calculation provided by the kit. Thus, finally the IDH activity was reported as nmol/min/μg = mU/μg. One unit of IDH is defined as the enzyme activity that will generate 1.0 μmol of NADH or NADP per minute at pH 8.0 and 37 °C. The enzyme activity was determined using Eq. (1) where B is the amount of NADH generated between end and the start of the reaction.

$$IDH\ activity = \frac{B}{reaction\ time \times \text{μg of protein}} \quad (1)$$

## Atomistic molecular dynamics simulations

The initial coordinates for the human IDH3α/γ dimer with $Mg^{2+}$, citrate and ADP were taken from the PDB (PDB ID: 5GRE[29]). Missing residues were fixed with MODELLER[63] and citrate parameters were generated with the CHARMM general force field (CGenFF) v2.4.0 using the ParamChem web server. Two systems were modeled: the IDH3α/γ dimer with $Mg^{2+}$, citrate and ADP and the same system with a disulfide-bond between Cys284 and Cys148 from the γ subunit. The protein complexes were set up in a truncated octahedral box containing water and NaCl ions accounting for a concentration of ~20 mM mimicking the mitochondrial matrix milieu. Five replicas of each system were run for 1 μs each using the CHARMM36m[64] force field in GROMACS v2019.3[65]. Each system was minimized, heated to 300 K, and equilibrated under NPT conditions using a Nosé-Hoover thermostat[66,67], and a Parrinello-Rahman barostat with a compressibility of $4.5\,10^{-5}\,bar^{-1}$ to a reference pressure of 1 bar. The integration timestep was set to 2 fs, and bonds to hydrogens were constrained with the LINCS algorithm. Long-range electrostatics were computed using particle-mesh Ewald with a mesh size of 0.12 nm; van der Waals interactions were shifted smoothly to zero between 1.0 nm and 1.2 nm.

RMSD analysis was performed with the gmx rms tool of Gromacs, and the force distribution analysis (FDA) with the FDA Gromacs tool (v2020)[31] to identify residues that experience differential forces due to the presence of the disulfide bond. In each of the independent simulations, the residue-based pairwise forces ($F_{ji}$, where $j$ and $i$ are residues) were calculated and the punctual stress ($S_i = \sum_j |F_{ji}|$) averaged over 10 blocks of the trajectory, resulting in total in 50 data points for each setup. To obtain the difference between the systems without (noSS) and with disulfide-bridge (SS), we computed the difference between their averages. Student's $t$-test was performed on the two series with 50 datapoints to pinpoint those residues that experience a statistically different force. Residues with a $p$-value below 0.0001 are highlighted in the dimer structure. Statistical tests were performed with scipy's statistical module (scipy.stats v1.5.4). Visualization was performed with VMD v1.9.4a51. For clarity, residues are numbered with the UNIPROT numbering followed by the subunit they belong to.

## Statistics

Statistical analyses for the in vivo and in vitro experiments were performed using GraphPad Prism software 9.0.0 (GraphPad Software). Two-way ANOVA with Tukey's test was used for the time-course echocardiography analysis and the isometric force measuremnts when comparing wt and HyPer-DAO mice treated with D-ala or L-ala. One-way ANOVA with Dunett's test was used for the time course echocardiography analysis when analysing the HyPer-DAO mice in the on off experiments. Two tailed one-sample t-test was used for the analysis of the fold change of RNA levels. One-way ANOVA with Bonferroni's test were used to compare enzyme activities, amount of PEGylated IDH3γ, ATP levels and relative levels of metabolites. Specific tests were identified in the respective figures. $p$-values and sample size can be found in the figures and the figure legends.

## Reporting summary

Further information on research design is available in the Nature Portfolio Reporting Summary linked to this article.

## Data availability

The mouse line generated in this study will be deposited to the Knockout Mouse Project (KOMP). The proteomics data generated in this study have been deposited in the ProteomeXchange Consortium via the PRIDE partner repository under accession code PXD037987. All other data generated in this study are provided in the Supplementary Information/Source Data file. Source data are provided with this paper.

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

## Acknowledgements

This work was funded by the Deutsche Forschungsgemeinschaft (DFG) IRTG1816 and Ka 1269/13-1 to D.M.K., by Germany's Excellence Strategy

—EXC 2067/1-390729940 to P.R., T.B. and the DFG-funded SFB1002 (project A06 to P.R., A09 to S.B. and A14 to T.B.). We thank Annette Hillemann and Katja Brechtel-Curth for their expert technical support. We acknowledge support by the Open Access Publication Funds of the Göttingen University.

## Author contributions

Experimentation, M.S.N., A.M.V.L., J.G., G.J.v.B., A.J., E.S.P., A.F.B., R.S., A.Z., C.M.S., K.D., S.B., M.S., J.L., H.W., S.B., A.C.; analysis, M.S.N., A.M.V.L., J.G., A.F.B., R.A.B., K.D., C.F., J.B., I.B., J.R., P.R., T.B., V.V.B., D.M.K.; manuscript preparation M.S.N., A.M.V.L., J.G., A.F.B., P.R., V.V.B. and D.M.K.; supervision V.V.B. and D.M.K.; funding acquisition D.M.K. All authors reviewed the manuscript.

## Funding

## Competing interests

The authors declare no competing interests.

## Additional information

[1]Institute of Cardiovascular Physiology, University Medical Center Göttingen, Georg-August, University Göttingen, 37073 Göttingen, Germany. [2]Computational Biology, Friedrich-Alexander-Universität Erlangen-Nürnberg, 91058 Erlangen, Germany. [3]Erlangen National High-Performance Computing Center (NHR@FAU), Erlangen, Germany. [4]Institute of Functional Genomics, University of Regensburg, 93053 Regensburg, Germany. [5]University of Cologne, Faculty of Medicine and University Hospital of Cologne, Cluster of Excellence Cellular Stress Responses in Aging-associated Diseases (CECAD), 50931 Cologne, Germany. [6]Department of Internal Medicine, Section on Molecular Medicine, Wake Forest University School of Medicine, Winston-Salem, NC 27157, USA. [7]Clinic of Cardiology & Pneumology, University Medical Center Göttingen, Göttingen, Germany. [8]DZHK (German Center for Cardiovascular Research), Partner Site Göttingen, Göttingen, Germany. [9]King's College London, School of Cardiovascular Medicine & Sciences, The British Heart Foundation Centre of Excellence, SE1 7EH London, UK. [10]Institute for Biochemistry, Redox Metabolism and CECAD, University of Cologne, 50674 Cologne, Germany. [11]Institute of Cellular Biochemistry, University Medical Center Göttingen, 37073 Göttingen, Germany. [12]Cluster of Excellence, Multiscale Bioimaging: from Molecular Machines to Networks of Excitable Cells (MBExC), University of Göttingen, Göttingen, Germany. [13]Fraunhofer Institute for Translational Medicine and Pharmacology ITMP, Translational Neuroinflammation and Automated Microscopy, Göttingen, Germany. [14]Max Planck Institute for Multidisciplinary Sciences, 37077 Göttingen, Germany. [15]Federal Center of Brain Research and Neurotechnologies, Federal Medical Agency, 117997 Moscow, Russia. [16]Present address: Department of Biochemistry, University of Oxford, Oxford OX1 3QU, UK. ✉e-mail: doerthe.katschinski@med.uni-goettingen.de

