## [Peer Review File · Nature Communications]

IDH3 γ functions as a redox switch regulating mitochondrial energy metabolism and contractility in the heartREVIEWER COMMENTS

Reviewer #1 (Remarks to the Author):

In their study Nanadikar and colleagues apply a genetically encoded tool for H₂O₂ production recently developed by the Belousov group (also part of this study) to study H₂O₂ linked redox metabolism in heart. By overproducing H₂O₂ in mice heart they identify specific cysteine redox switch in IDH3 that allows cells to directly sense cellular redox state. Presented work is innovative as it is one of the first examples when both a genetically encoded sensor and tool were expressed as one construct. This allows authors to both generate H₂O₂ and at the same time to directly monitor live changes in H₂O₂ levels. Overall, this study represents an important advancement in our understanding how cells sense changes in redox environment and how they respond to them via a Cys-based redox switch. This study also illustrates usefulness of the Oximouse project recently introduced by the Chouchani group. Although this study presents an advancement towards our understanding of redox regulation in live cells, the presented work needs to be improved to be considered for a publication. Below are my specific comments:

- 1). It looks like the H₂O₂ levels produced by HyPerDAO is rather modest both based on a very high K_M for D-Ala as well as on physiological effects presented. This mild H₂O₂ production perhaps represents an advantage compared to other methods of H₂O₂ generation in cellular or organismal systems. It is important to see where H₂O₂ produced by HyPerDAO lies compared to other methods of H₂O₂ production (i.e. addition of menadione or other redox cyclers that via NQO1 mediate H₂O₂ production; H₂O₂ levels due to inhibited ETC, bolus of H₂O₂ directly added to cells etc).
- 2). How the HyPerDAO construct was developed in first place? Are there any linker regions between HyPer and DAO proteins? Was HyPerDAO protein previously expressed in E.coli and purified to show that the enzymatic activity of HyPerDAO is very similar to DAO alone?
- 3). The schematic of the DAO reaction in Fig. 1A is not correct: D-ala conversion leads to imino acid formation and FAD reduction (FADH₂), then FADH₂ reacts with O₂ to produce H₂O₂. What happens next to the imino acid product? Will imino acid react with H₂O with a concomitant NH₃ production in cells? Were ammonia levels evaluated in DAO expressing cells/organs?
- 4). Not clear the reasoning why nuclear targeted HyPerDAO construct was used in mice. It seems logical to use a cyto construct (it looks like mitochondria are very good at detoxifying H₂O₂ (especially low levels produced by DAO?) and not usage of a mito construct in mice was well justified).
- 5) In Fig. 1 B-C which antibody was used to detect HyPerDAO? Loading GAPDH control was not equal in samples presented.
- 6). Fig. 2: Blue colors are very confusing, especially half blue bars with a gradient, therefore representations need to be improved to help with the clarity.

7). Why only IDH3 was explored as an enzyme under redox control? It seems like SHMT1 enzyme is also worth exploring given its role in NAD(P)H and 1C metabolism.

8). It is very difficult to gauge enzymatic activities results as in Methods only a kit from Sigma was mentioned. How exactly these assays were performed, which substrates, under Vmax conditions etc? Were cell homogenates desalted to remove endogenous substrates? This study will greatly benefit from tracer studies (at least in a cell line model presented). This will allow to directly evaluate glycolysis/TCA cycle activities and fluxes. Also, a summary schematic figure is needed to guide the reader on the state of redox Cys and downstream effects linked with the IDH3 activity. Obviously even mild H₂O₂ production by DAO leads to downstream effects in cells (including decreased ATP levels) and it's not expected that mutated redox Cys will rescue ATP production to a full extent. At the same time a connection between an increased TCA cycle activity (via IDH3) and GSH biosynthesis is not very convincing. At least in HEK293T cells NADH/NAD⁺ and NADPH/NADP⁺ levels need to be evaluated; what happens if cells are treated with buthionine sulfoximine (to inhibit GSH biosynthesis).

9). It is certainly will be more convincing to see the effect observed not only in HEK293T cells. Have authors tried another cell line? For example, C2C12 myoblasts? It will be very interesting if in a panel of cell lines IDH3 activity will be increased upon DAO-mediated H₂O₂ production.

10). As mentioned above a schematic figure is needed to guide the reader through results: the state of redox cysteine switch vs TCA activity and other downstream effects;

Reviewer #2 (Remarks to the Author):

Nandikar, et. al, propose that IDH3 γ is a redox switch responsible for losses of cardiac function under significant ROS-induced stress. The research design is of notable robustness, and the authors present a fairly convincing justification for this mechanism as a possible source of myocardial "stunning" after infarction. While the overall manuscript is strong, some parts of the results and discussion could be refined to make the message clearer.

p.6. lines 187-189. The authors state cytosolic H₂O₂ diffuses into the mitochondria, but not vice-versa. This by definition is not diffusion. Should it be termed "facilitated transport by an unknown mechanism?"

p.9 line 301. The oxygen consumption rate (OCR) is the essential element indicative of increased TCA cycle turnover. But that data is not presented (see (2) below). Instead, the authors convolve increased metabolite pool sizes with increased activity, which is plausible, but does not establish a true link. See reference:" Ragavan, Mukundan, et al. "A comprehensive analysis of myocardial substrate preference emphasizes the need for a synchronized fluxomic/metabolomic research design." *American Journal of Physiology-Heart and Circulatory Physiology* 312.6 (2017): H1215-H1223."

With the interplay between energy metabolism at IDH3 and glutathione availability, it is quite possible that changes in pool sizes are related to anaplerotic flux into the TCA cycle. "Brunengraber, Henri, and Charles R. Roe. "Anaplerotic molecules: current and future." *Journal of inherited metabolic disease* 29.2 (2006): 327-331."

A secondary question is that of IDH3 reversibility. While IDH1 and IDH2 are known to be reversible, IDH3 is not generally acknowledged to have the same properties. Did the authors find any evidence of reversibility in this context? That would add to the originality of this manuscript.

Finally, the connection simply was not clear to me between increased IDH3 function and loss of ATP production (lines 290-292). If IDH is rate limiting for TCA cycle flux as stated in the document, an increase should mean more NADH, and more ATP production, not less. Maybe I am missing the logic, but I believe the discussion could be made clearer. Lines 380-384 draw a connection between increased TCA cycle flux and glutamine cataplerosis, but simultaneously invokes glutamine anaplerosis. Please clarify.

Data to be presented:

- 1) The weights of the control versus HyPer hearts would be nice to include along with the histology of Figure 1.
- 2) The raw SeaHorse data should be included in the main figures, with a full description of the data. The actual oxygen consumption rates need to be included in a clear manner. While Figure 7E ostensibly refers to the OCR, the caption for this figure is abbreviated. Is the fluorescence on y-axis associated with OCR?

Reviewer #3 (Remarks to the Author):

This is an intriguing work identifying isocitrate dehydrogenase 3 (IDH3) as a metabolic enzyme that can act as a redox switch through reversible oxidation, alongside important insights on the metabolic consequences of elevated H₂O₂ in the heart. IDH3 oxidation was identified through the use of an aMHC-HyPer-DAO mouse model, which is an innovative in vivo system that affords the cardiomyocyte overexpression model of H₂O₂ as well as an H₂O₂ reporter in the nucleus. The overall findings that IDH3 can be reversibly redox modified and the metabolic consequences of this modification are significant and the approach was innovative. However, the physiological significance of these findings are less clear. Mitochondria are a major source of ROS/H₂O₂, whereas the nucleus is not often described as such. Accordingly, it is unclear why the authors chose to model in vivo elevated H₂O₂ signaling in the nuclear compartment when they had access to the mitochondrial construct as evidenced by use in vitro studies. Moreover, while the aMHC-HyPer-DAO mice do show that IDH3 modification can occur in the heart, does this happen physiologically, or in the context of cardiac disease? While the authors do note in the discussion that oxidation may play a particularly important role in acute myocardial

infarction/myocardial stunning, ROS signaling may also be important in other disease modalities such as cardiac pressure overload. Showing IDH3 redox modification in a physiologically relevant cardiac disease model would strengthen the study and expand the disease relevance of this work.

Concerns:

1. Using the aMHC-HyPer-DAO-NLS mice to generate a gradient of H₂O₂ is intriguing, however, the rationale underlying the choice to use the NLS HyPer-DAO to make the mouse model is not clear. Mitochondria are a major source of ROS in cardiomyocytes and in cardiac disease. The explanation for this choice, especially in light of the availability of the mitochondrially targeted construct to the authors, needs to be further expanded upon. Is the concentration of H₂O₂ produced in vivo with the NLS-targeted construct indeed higher in the nucleus than in the mitochondria? Also how far does H₂O₂ travel within the cell? Does this system reflect the levels of H₂O₂ that would be produced in a cardiac disease setting? Please include in the discussion/rationale for NLS HyPer-DAO choice.
2. In Figure 1E, the authors show serial H&E stained sections of 2 hearts HyPer-DAO vs WT to show that the hearts are unaffected with HyPer-DAO expression at baseline. These images are not informative— please include echocardiography of baseline cardiac function and morphometry to accompany these images in the supplemental data.
3. In Figure 2C, the authors show that over the course of 21 days, activation of the cardiomyocyte HyPer-DAO construct can result in reduced fractional shortening and ejection fraction) without adverse cardiac wall remodeling. Moreover, the authors show in 2D that this contractile dysfunction can be reversible. These findings are striking, and should be accompanied by the echocardiography measurements of left ventricular interior dimensions at diastole and systole, as well as the stroke volume and end diastolic volumes.
4. Are the gene expression changes observed with HyPer-DAO activation (Fig. 2A) also reversible/inducible with the removal of D-alanine (in line with the changes in cardiac function shown in the dosing regimen used in 2D)?
5. Are the mitochondrial proteins identified by mass spec to be reversibly modified in Fig3B also modified with HyPer-DAO constructs targeted to other cellular compartments?
6. The aMHC-HyPer-DAO-NLS animal is a really great tool to highlight what can be redox modified in the heart, but there is a disconnect between what is identified with this nuclear H₂O₂ source model, versus what might occur physiologically in the heart (where the mitochondria might be the major source of ROS). This study would be very much strengthened if the authors would be able to show that some of the redox modified targets identified in the HyPer-DAO mice to be similarly modified in cardiac disease settings (would be particularly of interest to know if IDH3 Cys148 plays a role). While the authors note in the discussion that this may play a role in myocardial stunning, it would also be worthwhile to look in the context of TAC, where ROS may as play a role (also because the authors already have that surgical model in use for this study).

Reviewer #4 (Remarks to the Author):

With the HyPer-DAO mouse model and redox proteomics, Nanadikar et al. identified IDH3 γ that can serve as a redox switch, revealing that the Cys148 and Cys284 are the two critical sites in regulating IDH3 activity. The experiment design and the results are generally sound. The finding on the redox regulatory role of IDH3 γ is interesting. The authors should address the following questions before it becomes acceptable.

1. The in vivo H₂O₂ abundance in mouse heart with and without D-ala treatment was not determined. How much was the basal H₂O₂ in mouse heart? and how much H₂O₂ was produced in response to D-ala?
2. Do ATP production and other energy-related metabolites differ between HyPer-DAO and WT with the D-ala treatment in mice?
3. Fig 4EG. The PEG switch experiment shows that the oxidative level of IDH3 is low. Can this minor redox level change result in sufficient regulation on total enzyme activity?
4. Fig 7C. Why was PEGylated IDH3 much higher in C-284A mutant? Should PEGylation level be correlated to its enzymatic activity? And why? It looks like the lane image of the C284A treating with D-ala was cropped from elsewhere. Please confirm this is the original image.
5. Since this is a cys-site specific analysis, the redox proteomics data should be analyzed and illustrated in a site-centric manner. The current data (e.g. fig3b and SI tables) analyses were all performed on protein level. An average value of all cys sites surely dilutes the importance of those proteins in which only a part of the cys sites show changes. I would strongly suggest the authors to re-analyzed their proteomic data in a site-centric manner.
6. The MS2 and fold-changes of the important cys sites, such as cys 148 and 284 of IDH3 γ , should be added in the manuscript. Their identification with high-confidence is the foundation of follow-up studies.
7. The STN is a very weird term in describing fold-change in proteomics. Signal-to-noise has an exact definition in mass spectrometry, but rarely represents fold-change. The authors should revise this word, and related figures and tables.

8. The overall quality of the redox proteomics should be assessed. In this reviewer's opinion, there are several flaws.

a. Why was the enrichment done on proteins, not on peptides? A peptide-based enrichment would deepen the analysis coverage.

b. How much was the enrichment efficiency (cys-containing peptides vs. total peptides)? Based on several example raw data the authors provided, this reviewer found that the enrichment efficiency was bad. Roughly, only less than 10% of identified peptides contain cys. This is a sign that the experiments were not performed properly. Moreover, in the several data files I searched, the cys 148 and 284 of IDH3 γ were not identified.

c. Total numbers of cys sites and changed cys sites should be added. Their quantitative results should be included.

d. Blocking free cys with MSTP was done in a RIPA buffer, which was a mild buffer thus couldn't denature all proteins' structures. Therefore, some free cys sites might not be blocked in the first step, affecting the subsequent enrichment. The authors should evaluate the completeness of the reaction between free cys and MSTP.

9. Line 107. How does Nrf2 itself change?

10. Line 169 and fig 3e. The RNA levels of IDH3 were increased, though the protein level was not. Is the difference statistically significant?

11. Line 171 and fig 3G. Does IDH3 activity really go back to normal after removing D-ala for 2days? Is the difference between the two bars on the right statistically significant?

12. Fig 7B. the IDH3 basal activity of C236A is lower than wt IDH3 and other mutants. Are there any explanations?

Reviewer comments - Point to Point response

We would like to thank all four reviewers for their input. In response to their comments we have performed the following new experiments. A more detailed point to point response can be seen below:

1. Echocardiography analysis of cardiac output, stroke volume, left ventricular inner diameter in systole, left ventricular inner diameter in diastole, enddiastolic volume at basal level and after D-ala or L-ala treatment (Fig. 1E, Suppl. Fig. 1B).
2. Quantification of Nrf2 target RNAs in left ventricular samples obtained from wt and HyPer-DAO mice after 7 days of treatment with D-ala plus 2 additional treatment free days (Fig. 2A).
3. Analysis of heart weight to body weight of wt and HyPer-DAO mice before and after D-ala treatment (Fig. 2E).
4. Analysis of ATP/ADP levels in left ventricular samples of wt and HyPer-DAO mice after a 7 days long treatment with D-ala (Fig. 2F).
5. Repetition of the redox proteomics screen with left ventricular samples from wt and HyPer-DAO mice after 7 days of treatment with D-ala as well as a peptide centric analysis (Fig. 3B and C, Suppl. Table 1 and 2)
6. IDH3 activity and IDH3 redox modification analyzed in heart samples obtained from mice after transverse aortic constriction or sham treatment (Fig. 3J and K).
7. IDH3 redox modification analyzed in HEK HyPer-DAO NLS, HyPer-DAO MLS and HyPer-DAO NES cells by Peg switch assays (Suppl. Fig. 3B).
8. IDH3 activity in a panel of wt cells after H₂O₂ bolus treatment as well as in HyPer-DAO NLS or HyPer-DAO NES stably transfected Hep3B cells after D-ala treatment (Suppl. Fig. 3D).
9. Targeted stable isotope tracer studies using uniformly labeled ¹³C glucose (Fig. 7F and G).
10. NADH/NAD⁺ and NADPH/NADP⁺ levels in HEK HyPer-DAO NLS cells after L- or D-ala treatment (Fig. 7H).
11. ATP levels after treatment of HEK HyPer-DAO cells with buthionine sulfoximine (see rebuttal letter, reviewer #1, comment #8.3).
12. HyPer fluorescence measurements in HEK HyPer-DAO NLS, MLS and NES cells after treatment with menandione, rotenone or antimycin A (see rebuttal letter, reviewer #1, comment #1)
13. Nrf2 Western blot with protein extracts of wt and HyPer-DAO mice after a 7 days long treatment with D-ala (see rebuttal letter, reviewer #4, comment #9).
14. IDH1/2 activities, IDH3 activity and ATP levels in HEK HyPer-DAO cells harboring Cys148A/Cys284A double point mutations (Suppl Fig. 5A-C).

Reviewer #1 (Remarks to the Author):

Changes made in response to reviewer #1 are marked in **turquoise** (changes made in response to >1 reviewer are marked in **grey**) in the revised manuscript.

In their study Nanadikar and colleagues apply a genetically encoded tool for H₂O₂ production recently developed by the Belousov group (also part of this study) to study H₂O₂ linked redox metabolism in heart. By overproducing H₂O₂ in mice heart they identify specific cysteine redox switch in IDH3 that allows cells to directly sense cellular redox state. Presented work is innovative as it is one of the first examples when both a genetically encoded sensor and tool were expressed as one construct. This allows authors to both generate H₂O₂ and at the same time to directly monitor live changes in H₂O₂ levels. Overall, this study represents an important advancement in our understanding how cells sense changes in redox environment and how they respond to them via a Cys-based redox switch. This study also illustrates usefulness of the Oximouse project recently introduced by the Chouchani group. Although this study presents an advancement towards our understanding of redox regulation in live cells, the presented work needs to be improved to be considered for a publication. Below are my specific comments:

Answer: We would like to thank reviewer #1 for the in general positive evaluation of our manuscript. Please find the answers to the specific comments below.

1). It looks like the H₂O₂ levels produced by HyPerDAO is rather modest both based on a very high KM for D-Ala as well as on physiological effects presented. This mild H₂O₂ production perhaps represents an advantage compared to other methods of H₂O₂ generation in cellular or organismal systems. It is important to see where H₂O₂ produced by HyPerDAO lies compared to other methods of H₂O₂ production (i.e. addition of menadione or other redox cyclers that via NQO1 mediate H₂O₂ production; H₂O₂ levels due to inhibited ETC, bolus of H₂O₂ directly added to cells etc).

Answer: Monitoring H₂O₂ production using HyPer provides some clues regarding the concentrations of H₂O₂ produced under different concentrations of D-ala. In the original paper, in which the biosensor HyPer was described, *in vitro* calibrations of purified HyPer towards different concentrations of H₂O₂ were reported (Belousov et al, Nat Meth 2006, doi: 10.1038/nmeth866). The minimal probe response was elicited to be around 25 nM, and complete oxidation to be around 200 nM H₂O₂. The basal concentration of H₂O₂ in the cytosol is in the low nanomolar range (Belousov, Oxidative Stress 2020, Chapter 6, doi.org/10.1016/B978-0-12-818606-0.00006-7). Therefore, HyPer indeed allows to detect a range of concentrations from basal level up to 1-2 orders of magnitude higher compared to the basal state, which can be interpreted by the cell as a signal or a stress, depending on the duration.

There are examples how DAO-driven H₂O₂ production can be compared to other ways of oxidative challenge. First, we frequently use bolus H₂O₂ addition to the cells at the end of a DAO experiment to determine saturation of the probe (example in Pak et al, 2020, Cell Metab,

doi: 10.1016/j.cmet.2020.02.003). Bolus addition of H₂O₂ was also applied in the current manuscript (see Fig. 4B). Typically, for bolus addition saturating concentrations of H₂O₂ are used. The main differences of H₂O₂ bolus treatment compared to DAO are the following. First, bolus addition of moderate amount gives a short transient of H₂O₂ because of the rapid degradation of the oxidant, compared to the steady production of H₂O₂ by DAO. Second, DAO can be targeted to a particular organelle or cell type, which allows to create various scenarios of endogenous H₂O₂ production to evaluate the consequences for the cell or tissue. In contrast, bolus H₂O₂, redox cyclers, or mitochondria poisons affect equally all cells in a tissue, or, in case of bolus H₂O₂, all compartments in the cell, which represents a less physiological scenario. In previous experiments from our group we used rotenone which resulted in much lower levels of H₂O₂ production compared to DAO, and only in the mitochondrial matrix, but not in the cytosol (see Fig 3F and G in Pak et al, 2020, Cell Metab, doi: 10.1016/j.cmet.2020.02.003). Moreover, the detection was possible with the highly sensitive HyPer7 and not with the earlier version HyPer3. In our current study we employed HyPer2, which is a similar version as HyPer3. Conceptually, giving the cell mitochondrial poisons is a disadvantaged way to induce oxidative stress, because together with poorly controlled and quantifiable H₂O₂ production this also shuts down ATP synthesis and, likely, affects many other functions of mitochondria. Instead, DAO can mimic a more or less "pure" oxidative stress or signaling depending on the enzyme location, substrate concentration, and exposure timing.

To further answer the question experimentally, we strictly followed the advice of reviewer #1, despite the above arguments, and compared the D-ala induced HyPer response with the response of cells to menadione and inhibition of the electron transport chain via treating the cells with antimycin A or rotenone.

The redox cyclers menadione but not inhibitors of the ETC induce a ROS response similar to D-ala detected by HyPer2. HEK HyPer-DAO-MLS, -NLS and -NES cells were treated either with 50 mM D-ala, 50 μ M menadione, 5 μ M rotenone or 5 μ M antimycin A. The substances were added at time point 4 min. Subsequently, HyPer2 fluorescence responses were recorded. Ratios are normalized to the HyPer ratio prior treatment, 40 cells per condition were analyzed. Mean \pm SEM.

As can be seen in these new experiments, the D-ala induced H₂O₂ responses in the nucleus, mitochondria and the cytoplasm induce a HyPer response as well as the treatment of the cells with 50 μM menandione. Inhibition of the electron transport chain with rotenone (complex I) or antimycin A (complex III), on the other hand, did not result in a response, which we were able to detect with the HyPer2 sensor, which is in line with the above described previous data from the Belousov laboratory (Pak et al, 2020, Cell Metab, doi: 10.1016/j.cmet.2020.02.003). Based on the above-mentioned arguments, we decided not including this data set in the manuscript, however, if advised by the reviewer to do so, we easily can show these data in the supplementary section.

2). How the HyPerDAO construct was developed in first place? Are there any linker regions between HyPer and DAO proteins? Was HyPerDAO protein previously expressed in E.coli and purified to show that the enzymatic activity of HyPerDAO is very similar to DAO alone?

Answer: HyPer-DAO fusion was the same as used before (Matlashov et al, 2014 Antioxid Redox Signal, doi: 10.1089/ars.2013.5618) and contained the Gly-Gly-Ser-Gly linker between HyPer and DAO. This information has been added in the Material and Method section (lines 402-403 of the revised manuscript).

Our earlier attempts to purify DAO failed, likely because E. coli have a substantial amount of some D-amino acids that induce oxidative toxicity (mutant inactive DAO version can be efficiently expressed). Instead, in the previous paper from the Belousov laboratory that describes HyPerDAO fusion (Matlashov et al, 2014, Antioxid Redox Signal, doi: 10.1089/ars.2013.5618), the authors compared HyPerDAO with DAO co-expressed with HyPer, and found that the enzyme retains its full functionality in both fused and free forms. HyPer fused to DAO appeared to be an even better detector of H₂O₂ produced by DAO compared to co-expressed version, obviously because in the fusion the generator and detector are placed in proximity to each other.

3). The schematic of the DAO reaction in Fig. 1A is not correct: D-ala conversion leads to imino acid formation and FAD reduction (FADH₂), then FADH₂ reacts with O₂ to produce H₂O₂. What happens next to the imino acid product? Will imino acid react with H₂O with a concomitant NH₃ production in cells? Were ammonia levels evaluated in DAO expressing cells/organs?

Answer: We apologize for the mistake made in Fig. 1A, which has been corrected in the revised manuscript (see Fig 1A of the revised manuscript). We also would like to point out that the presented graph is a simplified scheme of DAO reaction rather than the complete reaction scheme. Indeed, D-amino acids are dehydrogenated by DAO into imino acids that spontaneously hydrolyze to the corresponding α-keto acids and ammonia. In case of D-ala used as a substrate in our study, the resulting α-keto acid is pyruvate.

However, as the reaction is slow, and H₂O₂ is produced in nanomolar to low micromolar amounts, both ammonia and pyruvate are produced in equimolar amounts, several orders of magnitude lower than high-micromolar endogenous ammonia or millimolar pyruvate concentrations. Therefore, we assume that neither ammonia nor pyruvate produced by DAO are able to perturb

cellular metabolism. In addition, extensive ammonia generation would lead to at least a transient alcalinization of the cellular compartment of interest. Our numerous control experiments in many cell types expressing DAO with SypHer family pH probes did not show any alterations of cellular pH upon D-ala addition confirming that ammonia is produced at very low rate.

4). Not clear the reasoning why nuclear targeted HyPerDAO construct was used in mice. It seems logical to use a cyto construct (it looks like mitochondria are very good at detoxifying H₂O₂ (especially low levels produced by DAO?) and not usage of a mito construct in mice was well justified).

Answer: We indeed originally intended to generate aside from the nuclear targeted HyPer-DAO mice also transgenic mice that express HyPer-DAO in the mitochondrial matrix and the cytosol. We performed blastocyst injections twice. In the first round we obtained in total 5 HyPer-DAO NLS and 5 HyPer-DAO NES founder mice carrying the correct genotype. Functional analysis, however revealed that the cardiomyocytes of just one of the HyPer-DAO NLS mice showed a positive expression and D-ala response. In the other mice the HyPer-DAO fusion protein expression was either negligible, dislocated or completely degraded. In the second round of blastocyst injections we obtained 3 HyPer-DAO NES and 2 HyPer-DAO MLS mice genotyped positive by PCR. Cardiomyocytes of neither of them demonstrated a functional DAO-HyPer response due to the same reason as for the previous founder mice. We added this information to the revised manuscript in the result (see lines 91-93 of the revised manuscript) and in the discussion (lines 324-334 of the revised manuscript).

Although the generation of HyPer-DAO NES and HyPer-DAO MLS transgenic mice was not successful, we still believe that the data obtained with the HyPer-DAO NLS mice are of physiological relevance since (i) we verified the H₂O₂ effect on IDH3 γ redox modification (see new data set presented in Fig. Suppl. Fig. 3B) and IDH3 activity in Hek HyPer-DAO NLS, HyPer-DAO NES and HyPer-DAO MLS cells (see Fig. 4C) and more importantly (ii) we have in the meantime gained evidence that IDH3 activity is altered in a relevant cardiac disease setting (see Fig. 3J and K), i.e. cardiac hypertrophy/heart failure in consequence of an increased afterload. We have chosen this cardiac disease model based on comment #6 of reviewer #3. We discuss the importance of subcellular H₂O₂ production and especially the H₂O₂ buffering capacity of mitochondria as mentioned by the reviewer (see lines 324-330 of the revised manuscript).

5) In Fig. 1 B-C which antibody was used to detect HyPerDAO? Loading GAPDH control was not equal in samples presented.

Answer: HyPer-DAO was recognized using the anti-GFP antibody (11814460001, Sigma-Aldrich, line 620 of the revised manuscript). The antibody binds to the HyPer part of the fusion protein.

For each organ sample we indeed loaded the same amount of total protein. It is well described in literature that GAPDH expression varies by 15-fold between different tissue types, which is also reflected in Fig. 1C (Barber et al, 2005, *Physiol Genomics*, doi.org/10.1152/physiolgenomics.00025.2005). In the experiments shown, we aimed for detecting if HyPer-DAO is present in the transgenic mice and heart only and not in wt or other

organs, which can be seen in Fig. 1B and C. However, we agree with the reviewer that an absolute quantification of HyPer-DAO protein levels standardized by GAPDH protein levels would be precluded but was also not intended in the presented manuscript.

6). *Fig. 2: Blue colors are very confusing, especially half blue bars with a gradient, therefore representations need to be improved to help with the clarity.*

Answer: The blue color in Fig. 2 was removed as suggested by the reviewer (see Fig. 2).

7). *Why only IDH3 was explored as an enzyme under redox control? It seems like SHMT1 enzyme is also worth exploring given its role in NAD(P)H and 1C metabolism.*

Answer: We definitively agree with reviewer #1 that IDH3 γ is not the only interesting candidate that was identified in the redox proteomics screen. We indeed followed up possible changes in the activity of the succinate dehydrogenase, which we could exclude in the mouse and the cell model (Fig. 3E and Suppl. Fig. 3)

SHMT1, which is involved in NADPH and 1C metabolism as well as Trifunctional protein (TFP) β , which is involved in β -oxidation might link redox signaling and alterations in metabolism. Since, however, the respective necessarily follow up experiments are of different nature depending on the protein candidate to be studied, we had to make a choice for the current project. We will follow up on the other candidates, which to our opinion is out of the scope of the current manuscript. We discuss in the revised manuscript that other redox-modified target proteins might add to the phenotype observed in the Hyper-DAO mice (see lines 351-354 and lines 358-363 of the revised manuscript).

8). *It is very difficult to gauge enzymatic activities results as in Methods only a kit from Sigma was mentioned. How exactly these assays were performed, which substrates, under Vmax conditions etc? Were cell homogenates desalted to remove endogenous substrates?*

Answer: More detailed information on the IDH activity measurements are given in the revised manuscript (see lines 727-751 of the revised manuscript).

This study will greatly benefit from tracer studies (at least in a cell line model presented). This will allow to directly evaluate glycolysis/TCA cycle activities and fluxes.

Answer: We have performed targeted stable isotope tracer studies using uniformly ^{13}C glucose and GC-MS with the HEK HyPer-DAO NLS wt and HEK HyPer-DAO NLS C148A cells. The new data are presented in Fig. 7F and G.

Also, a summary schematic figure is needed to guide the reader on the state of redox Cys and downstream effects linked with the IDH3 activity.

Answer: A summary schematic figure has been added in Fig. 7 (see Fig. 7I).

Obviously even mild H_2O_2 production by DAO leads to downstream effects in cells (including decreased ATP levels) and it's not expected that mutated redox Cys will rescue ATP production to a full extent. At the same time a connection between an increased TCA cycle activity (via IDH3) and GSH biosynthesis is not very convincing. At least in HEK293T cells NADH/NAD⁺ and NADPH/NADP⁺ levels need to be evaluated.

Answer: A new data set describing NADH/NAD⁺ and NADPH/NADP⁺ levels in the HEK HyPer-DAO wild type and C148A gene edited cells are presented in Fig. 7H.

What happens if cells are treated with buthionine sulfoximine (to inhibit GSH biosynthesis).

Answer: We would like to thank the reviewer for suggesting this experiment. We inhibited glutathione (GSH) biosynthesis by buthionine sulfoximine (BSO) treatment as recommended. Whereas inhibition of GSH biosynthesis rescues the decrease of ATP after D-ala stimulation in the HyPer-DAO-NLS HEK cells, this effect is not seen in the HyPer-DAO-NLS Cys148A HEK cells. These data support the notion that TCA metabolites, which are used for GSH biosynthesis after D-ala stimulation stay in the TCA cycle and are metabolized for energy production. However, since this is a very indirect measure, we decided to soften the conclusion regarding the glutathione synthesis in the manuscript.

Inhibition of glutathione synthesis rescues ATP levels. HEK HyPer-DAO-NLS wild type (wt) and HEK HyPer-DAO C148A cells were treated or not treated with 500 μ M buthionine sulfoximine (BSO) for 18 hrs. Subsequently ATP levels were measured with the ATP-red fluorescence sensor at three different conditions, i.e. non treated (n.t.), L-ala (50 mM, 20 min), and D-ala (50 mM, 20 min). 25 cells were analyzed per condition. mean, ** $p < 0.01$ by one-way ANOVA.

9). It is certainly will be more convincing to see the effect observed not only in HEK293T cells. Have authors tried another cell line? For example, C2C12 myoblasts? It will be very interesting if in a panel of cell lines IDH3 activity will be increased upon DAO-mediated H_2O_2 production.

Answer: We have analyzed IDH3 activity in a panel of cell lines after treating the cells with a bolus of H_2O_2 . We indeed find an increased activity of IDH3 in a panel of different cells (MADA-MB-231, Hep3B and C2C12). Additionally, we have included Hep3B cells that are stably expressing HyPer-DAO in the cytoplasm or the nucleus (Hep3B HyPer-DAO NES and Hep3B HyPer-DAO NLS) in our

analysis. Comparable to the effect seen in the HyPer-DAO mice and the HEK HyPer-DAO cells, IDH3 activity is increased after treating the cells with D-ala. The new data is shown in Suppl. Fig. 3D and described in lines 207-211 and lines 336-337 of the revised manuscript.

10). As mentioned above a schematic figure is needed to guide the reader through results: the state of redox cysteine switch vs TCA activity and other downstream effects;

Answer: A summary schematic figure has been added in Fig. 7 (see Fig. 7I).

Reviewer #2 (Remarks to the Author):

Changes made in response to reviewer #2 are marked in green (changes made in response to >1 reviewer are marked in grey) in the revised manuscript.

Nandikar, et. al, propose that IDH3 γ is a redox switch responsible for losses of cardiac function under significant ROS-induced stress. The research design is of notable robustness, and the authors present a fairly convincing justification for this mechanism as a possible source of myocardial “stunning” after infarction. While the overall manuscript is strong, some parts of the results and discussion could be refined to make the message clearer.

We would like to thank reviewer #2 for the in general positive evaluation of our manuscript. Please find the answers to the specific comments below.

p.6. lines 187-189. The authors state cytosolic H₂O₂ diffuses into the mitochondria, but not vice-versa. This by definition is not diffusion. Should it be termed “facilitated transport by an unknown mechanism?”

Answer: The underlying mechanisms are indeed not clear. Therefore, we believe that neither describing a pure diffusion or transport is supported by a sufficient amount of data in the literature. Therefore, we have changed the wording as suggested by the reviewer (see page 6, lines 188-190 of the revised manuscript).

p.9 line 301. The oxygen consumption rate (OCR) is the essential element indicative of increased TCA cycle turnover. But that data is not presented (see (2) below). Instead, the authors convolve increased metabolite pool sizes with increased activity, which is plausible, but does not establish a true link. See reference:” Ragavan, Mukundan, et al. "A comprehensive analysis of myocardial substrate preference emphasizes the need for a synchronized fluxomic/metabolomic research design." American Journal of Physiology-Heart and Circulatory Physiology 312.6 (2017): H1215-H1223.” With the interplay between energy metabolism at IDH3 and glutathione availability, it is quite possible that changes in pool sizes are related to anaplerotic flux into the TCA cycle. “Brunengraber, Henri, and Charles R. Roe. "Anaplerotic molecules: current and future." Journal of inherited metabolic disease 29.2 (2006): 327-331.”

A secondary question is that of IDH3 reversibility. While IDH1 and IDH2 are known to be reversible, IDH3 is not generally acknowledged to have the same properties. Did the authors find any evidence of reversibility in this context? That would add to the originality of this manuscript.

Answer: We agree with reviewer #2 that metabolic flux assays would give a better insight into the H₂O₂ induced metabolic changes. As also described above (see reviewer #1, comment #8) we have performed targeted stable isotope tracer studies using uniformly ¹³C glucose and GC-MS to obtain experimental insight into the alterations in metabolism aside from oxygen consumption rate measurements (see new data set presented in Fig. 7F and G). IDH3 is described to catalyze a non-reversible reaction. The data obtained in our experiments do not indicate that the enzymatic reaction *per se* changes upon redox-modification. However, we would like to point out that the redox-modification-induced increased enzyme activity of IDH3 is reversible upon release of the H₂O₂ stimulation (see Fig. 4F and G).

Finally, the connection simply was not clear to me between increased IDH3 function and loss of ATP production (lines 290-292). If IDH is rate limiting for TCA cycle flux as stated in the document, an increase should mean more NADH, and more ATP production, not less. Maybe I am missing the logic, but I believe the discussion could be made clearer. Lines 380-384 draw a connection between increased TCA cycle flux and glutamine cataplerosis, but simultaneously invokes glutamine anaplerosis. Please clarify.

Answer: In line with the new data set obtained with the tracer studies we have refined the discussion of the alterations in the TCA cycle flux (see page 10, lines 374-381 of the revised manuscript). As also discussed above we have softened the conclusion regarding the glutamine cataplerosis.

Data to be presented:

- 1) *The weights of the control versus HyPer hearts would be nice to include along with the histology of Figure 1.*

Answer: We have analyzed the heart weights of the wt and HyPer-DAO mice, which clearly show no change neither before nor after D-ala treatment. These data are included in Fig. 2E of the revised manuscript. Based on the comment made by reviewer #3, comment #2 we have removed the H&E sections as they were regarded as *non-informative*.

- 2) *The raw SeaHorse data should be included in the main figures, with a full description of the data. The actual oxygen consumption rates need to be included in a clear manner. While Figure 7E ostensibly refers to the OCR, the caption for this figure is abbreviated. Is the fluorescence on y-axis associated with OCR?*

Answer: We would like to politely disagree with the reviewer. Fig. 7E is correctly labelled as fluorescence for y-axis as it reflects the ATP measurements performed with the ATP-Red dye (Merck, SCT045) and not the seahorse measurements.

Reviewer #3 (Remarks to the Author):

Changes made in response to reviewer #3 are marked in **blue/green** (changes made in response to >1 reviewer are marked in **grey**) in the revised manuscript.

This is an intriguing work identifying isocitrate dehydrogenase 3 (IDH3) as a metabolic enzyme that can act as a redox switch through reversible oxidation, alongside important insights on the metabolic consequences of elevated H₂O₂ in the heart. IDH3 oxidation was identified through the use of an α MHC-HyPer-DAO mouse model, which is an innovative in vivo system that affords the cardiomyocyte overexpression model of H₂O₂ as well as an H₂O₂ reporter in the nucleus. The overall findings that IDH3 can be reversibly redox modified and the metabolic consequences of this modification are significant and the approach was innovative. However, the physiological significance of these findings in this study are less clear. Mitochondria are a major source of ROS/H₂O₂, whereas the nucleus is not often described as such. Accordingly, it is unclear why the authors chose to model in vivo elevated H₂O₂ signaling in the nuclear compartment when they had access to the mitochondrial construct as evidenced by use in vitro studies. Moreover, while the α MHC-HyPer-DAO mice do show that IDH3 modification can occur in the heart, does this happen physiologically, or in the context of cardiac disease? While the authors do note in the discussion that oxidation may play a particularly important role in acute myocardial infarction/myocardial stunning, ROS signaling may also be important in other disease modalities such as cardiac pressure overload. Showing IDH3 redox modification in a physiologically relevant cardiac disease model would strengthen the study and expand the disease relevance of this work.

We would like to thank reviewer #3 for the thoughtful comments. As you can also see from the specific responses below, we have especially concentrated on answering the questions about the general conclusions made with the HyPer-DAO localized to the nuclear compartment and to prove that the changes observed are of relevance in the context of cardiac diseases.

Concerns:

1. *Using the α MHC-HyPer-DAO-NLS mice to generate a gradient of H₂O₂ is intriguing, however, the rationale underlying the choice to use the NLS Hyper-DAO to make the mouse model is not clear. Mitochondria are a major source of ROS in cardiomyocytes and in cardiac disease. The explanation for this choice, especially in light of the availability of the mitochondrially targeted construct to the authors, needs to be further expanded upon.*

Answer: As described above (please see answer to reviewer #1, comment #4) we indeed intended to generate HyPer-DAO transgenic mice with the expression of the fusion protein in either the mitochondrial matrix, the cytosol or the nucleus, from which despite several blastocyst injections only the nuclear targeted version resulted in a successful mouse line. We have in the meantime obtained a data set that indicates that modification of IDH3 γ plays a role in afterload-induced cardiac hypertrophy/failure (see comment #6).

Is the concentration of H₂O₂ produced in vivo with the NLS-targeted construct indeed higher in the nucleus than in the mitochondria? Also how far does H₂O₂ travel within the cell?

Answer: We would like to refer the reviewer to the experiments shown in Suppl. Figure 2 of the manuscript. In these experiments, we transfected the Hek HyPer (Green) DAO-NES, -NLS and -MLS cells with HyPer-Red, which was localized to the mitochondrial matrix. These experiments indeed imply that the H₂O₂ levels in the nucleus are higher than in the mitochondria in the HyPer (Green) DAO-NLS cells indicating a cellular H₂O₂ gradient. Even more, when HyPer (Green) DAO was localized to the mitochondria (right panel of the figure), the same D-ala concentration resulted in less H₂O₂ levels compared to the nuclear localization, which indicates a high H₂O₂ buffering capacity of the mitochondria. Although we cannot formally answer the question how far H₂O₂ “travels” within the cells, overall the data clearly indicate intracellular H₂O₂ gradients. This is in line with the previous characterization of the HyPer-DAO system (Mishina et al, 2019, Antioxid Redox Signal, doi: 10.1089/ars.2018.7697, 2011, Mishina et al, 2011, Antioxid Redox Signal, doi: 10.1089/ars.2010.3539).

Does this system reflect the levels of H₂O₂ that would be produced in a cardiac disease setting? Please include in the discussion/rationale for NLS HyPer-DAO choice.

Answer: As suggested in comment #6 we have by now analyzed IDH3 activity in a well described mouse heart failure model, i.e. cardiac hypertrophy and failure induced by transverse aortic constriction.

A discussion/rationale for analyzing HyPer-DAO NLS is included in the revised manuscript (see page 9, lines 324-334 of the revised manuscript).

2. *In Figure 1E, the authors show serial H&E stained sections of 2 hearts HyPer-DAO vs WT to show that the hearts are unaffected with HyPer-DAO expression at baseline. These images are not informative-please include echocardiography of baseline cardiac function and morphometry to accompany these images in the supplemental data.*

Answer: We have performed echocardiography to analyze cardiac output in non-treated male and female wild type (wt) and HyPer-DAO mice. This new data set is presented in Fig. 1E. There are no significant differences between the wt and respective HyPer-DAO mice at baseline. We would additionally like to point out that other baseline characterization (including fractional area shortening/FAS, ejection fraction/EF, anterior wall thickness/AwTh and posterior wall thickness/PwTh) is already shown in the day 0 time points presented in Fig. 2C, 2D and Suppl. Fig. 1A of the manuscript. All animals were analyzed by echocardiography before L-ala or D-ala treatment. FAS, EF, AwTh and PwTh were unaltered in the transgenic mice compared to the age-matched wt littermates. As suggested we additionally analyzed stroke volume (SV), left ventricular inner diameter in systole (LVIDs), left ventricular inner diameter in diastole (LVIDd)

and enddiastolic volume (EDV). Neither SV, LVIDs, LVIDd nor EDV differ between wt and HyPer-DAO mice before treatment at day 0 (Suppl. Fig. 1B).

3. *In Figure 2C, the authors show that over the course of 21 days, activation of the cardiomyocyte HyPer-DAO construct can result in reduced fractional shortening and ejection fraction) without adverse cardiac wall remodeling. Moreover, the authors show in 2D that this contractile dysfunction can be reversible. These findings are striking, and should be accompanied by the echocardiography measurements of left ventricular interior dimensions at diastole and systole, as well as the stroke volume and end diastolic volumes.*

Answer: As suggested, we have analyzed left ventricular interior dimensions at diastole and systole, stroke volume and end diastolic volume. Data are presented in Suppl. Fig. 1B in the revised manuscript.

4. *Are the gene expression changes observed with HyPer-DAO activation (Fig. 2A) also reversible/inducible with the removal of D-alanine (in line with the changes in cardiac function shown in the dosing regimen used in 2D)?*

Answer: According to the reviewer comment we have analyzed the Nrf2 target genes shown in Fig. 2A in left ventricular samples of HyPer-DAO mice either treated with D-ala for 7 days in the drinking water (on) or 2 additional days without D-ala (off). Wild type (wt) littermates treated for 7 days with D-ala in the drinking water served as controls. We observed a mild increase of Nrf2 target genes in the HyPer-DAO mice after D-ala treatment, the RNA levels were no longer increased after removal of D-ala. The new data set is presented in Fig. 2A of the revised manuscript and described on page 4, lines 108-110 and page 10, line 365 of the revised manuscript.

5. *Are the mitochondrial proteins identified by mass spec to be reversibly modified in Fig3B also modified with HyPer-DAO constructs targeted to other cellular compartments?*

Answer: We would like to point out that we checked in HyPer-DAO-NLS, HyPer-DAO-NES as well as HyPer DAO-MLS cells for redox-induced changes in IDH3 activity (Fig. 4C). The changes observed were independent of the cellular compartment, in which H₂O₂ was produced. In response to the reviewer comment we also analyzed IDH3 γ redox modification in the cells with the HyPer-DAO construct in the nucleus, cytosol and mitochondria by performing Peg switch assays. In line with the IDH3 activity, IDH3 γ modification was altered in response to D-ala in all three cell lines (see new data set in Suppl. Fig. 3B).

Although it would be of course very interesting to repeat the redox proteomics screen aside from the hearts obtained from HyPer-DAO NLS mice (as demonstrated in the manuscript) in comparison with all three different cell lines, we believe that this would be an independent project and beyond the scope of the current manuscript. We hope for the understanding of the

reviewer that these kinds of experiments, although already planned, will not be included in the current project.

6. The α HMC-HyPer-DAO-NLS animal is a really great tool to highlight what can be redox modified in the heart, but there is a disconnect between what is identified with this nuclear H_2O_2 source model, versus what might occur physiologically in the heart (where the mitochondria might be the major source of ROS). This study would be very much strengthened if the authors would be able to show that some of the redox modified targets identified in the HyPer-DAO mice to be similarly modified in cardiac disease settings (would be particularly of interest to know if IDH3 Cys148 plays a role). While the authors note in the discussion that this may play a role in myocardial stunning, it would also be worthwhile to look in the context of TAC, where ROS may as play a role (also because the authors already have that surgical model in use for this study).

Answer: We would like to thank the reviewer for the advice. We analyzed left ventricular samples obtained from mice that underwent transverse aortic constriction (TAC) or sham surgery for IDH3 activity and IDH3 γ redox modification. As shown in Fig. 3J and K IDH3 activity and IDH3 γ modification are both increased in TAC challenged compared to sham treated mice. The data are described on page 6, lines 172-176 and discussed on page 11, lines 395-398.

We understand that it would be of particular interest to know if IDH3 γ Cys148 plays a role. To answer this question formally correct, the generation of a Cys148 point mutated mouse model would be necessary, which clearly is beyond the scope of the first description of IDH3 γ as a redox switch protein target.

Reviewer #4 (Remarks to the Author):

Changes made in response to reviewer #4 are marked in pink (changes made in response to >1 reviewer are marked in grey) in the revised manuscript.

With the HyPer-DAO mouse model and redox proteomics, Nanadikar et al. identified IDH3 γ that can serve as a redox switch, revealing that the Cys148 and Cys284 are the two critical sites in regulating IDH3 activity. The experiment design and the results are generally sound. The finding on the redox regulatory role of IDH3 γ is interesting. The authors should address the following questions before it becomes acceptable.

We would like to thank reviewer #4 for the in general positive evaluation of our manuscript. Please find the answers to the specific comments below.

- 1. The in vivo H_2O_2 abundance in mouse heart with and without D-ala treatment was not determined. How much was the basal H_2O_2 in mouse heart? and how much H_2O_2 was produced in response to D-ala?*

Answer: These are highly relevant but likewise difficult questions to be answered due to technical limitations. To quantify basal and D-ala induced H_2O_2 one would need to analyze the *in vivo* abundance of H_2O_2 in the mouse heart in the living animal optimally with unopened chest over days. Since the HyPer biosensor relies on its fluorescence characteristics, this setting would preclude detecting the HyPer signal in living mice. A real-time fiber-optic recording of fluorescence biosensor signals has been developed in the context of acute-ischemic-stroke in the brain recently (Pochechuev et al, 2022, J Biophotonics, doi: 10.1002/jbio.202200050). A similar system would be needed to be developed for the heart. Considering that the heart contains spontaneously beating musculature, developing a suitable fiber optics set up would need to account in addition these movements. Currently a suitable system does not exist in either the authors groups nor – to the best of our knowledge - somewhere else. Establishing a suitable setup is beyond the scope of the current project. Therefore, we hope for the understanding of the reviewer that solving these technical problems would be an independent project *per se*.

2. *Do ATP production and other energy-related metabolites differ between HyPer-DAO and WT with the D-ala treatment in mice?*

Answer: We agree with the reviewer that analyzing ATP in the hearts of the HyPer-DAO mice are of interest. To this end we determined the ATP/ADP ratio in wild type and HyPer-DAO mice after a 7day long treatment with D-ala (Fig. 2F). There is a trend of a decreased ATP-ADP ratio ($p = 0.08$) in the transgenic mice. This mild effect might be caused by the fact that aside from the affected HyPer-DAO expressing cardiomyocytes, the measurement will also include the ATP-ADP ratio of the non-cardiomyocytes. This conclusion is in part supported by the fact that we saw a significant decrease in ATP levels after D-ala treatment of cardiomyocytes isolated from HyPer-DAO mice (Fig. 5C).

3. *Fig 4EG. The PEG switch experiment shows that the oxidative level of IDH3 is low. Can this minor redox level change result in sufficient regulation on total enzyme activity?*

Answer: We certainly agree that a minor fraction of IDH3 γ was modified according to the PEG switch assay. This still resulted in a significantly altered and robust IDH3 activity. IDH3 is a rate limiting enzyme of the TCA cycle, with IDH3 γ as its regulatory component. Therefore, one might argue that an alteration of the entire IDH3 γ pool might result in drastic metabolic changes that would harm the cells. The IDH3 γ redox modification (PEG switch, Fig. 4G and G) and IDH3 activity (enzyme activity measurements, Fig. 4C and F) that we observed after D-ala stimulation were rapidly reversible, which might result in the correct titration that is needed for the on-off characteristic of the biological consequences. We have included a respective discussion on page 9, lines 335-336 of the revised manuscript.

4. *Fig 7C. Why was PEGylated IDH3 much higher in C-284A mutant? Should PEGylation level be correlated to its enzymatic activity? And why? It looks like the lane image of the C284A treating with D-ala was cropped from elsewhere. Please confirm this is the original image.*

Answer: Please find the uncropped image below. The wt and cysteine mutated cells were indeed analyzed side by side in the same experiment and immunoblot. Since the loading of the experiment performed with the C148A and the C284A mutant and the respective wild type (wt) cells were different (see original blots on the right side), the image was cut and rearranged (for the C148A mutant cells). We left a white space in the figure 7C, where the blot was cut and will make this space even clearer in the revised manuscript.

Fig 7C

In addition, it is absolutely correct that the PEGylation levels in the C284A mutant seem to be higher compared to the wt. We speculate that the reason why this band is more intense could be that the oxidation of cysteine 148 is more stable in the absence of cysteine 284. This could be explained if the disulfide is more solvent accessible than a cysteine 148 sulfenic acid.

5. *Since this is a cys-site specific analysis, the redox proteomics data should be analyzed and illustrated in a site-centric manner. The current data (e.g. fig3b and SI tables) analyses were all performed on protein level. An average value of all cys sites surely dilutes the importance of those proteins in which only a part of the cys sites show changes. I would strongly suggest the authors to re-analyzed their proteomic data in a site-centric manner.*

Answer: As a part of our major effort in this revision, we repeated the LC-MS/MS analysis of the samples employing a recently acquired Orbitrap Eclipse Tribrid mass spectrometer (Thermo Scientific, Waltham, MA), an instrument superior to the previously used Orbitrap Velos Pro mass spectrometer. The new proteomics and redox proteomics data (.raw files and database search results) were uploaded onto the ProteomeXchange via PRIDE server. Improvements in both overall protein and peptide numbers were noted as shown in the bar graphs below.

The analysis shows significant enrichment in cysteine-containing peptides from ~5% in the global proteomics data to >60% in the redox proteomics data. This demonstrates that the enrichment at the protein level in the redox proteomics workflow was effective to allow monitoring of redox changes. The new analysis shows a similar trend in differential protein abundance. Most importantly, the target protein, IDH3 γ was identified with the same changes in the Cys148 peptides abundance levels upon D-ala treatment. Peptide identification quality was acceptable judging from manual inspection of their MS2 spectra and XCorr scores (Sequest scores).

Proteomics data were summarized in peptides centered manner rather than protein comparison in the revised manuscript (see Supplement table 1 and 2). In the table, cysteine containing peptides in the 7 days treatment group was compared by the ratio of peak abundance calculated by label-free quantitation. A panel (as shown below) showing the number of unique non-cys and cys containing peptides and number of peptide spectrum matches (PSM) was added to Figure 3B to summarize the new data.

Venn-diagrams demonstrate elevated yield of cysteine-containing unique peptides and total number of identified peptide spectra matches (PTM) by redox enrichment.

The new redox proteomics data (.raw files and database search results) were uploaded onto the ProteomeXchange via PRIDE server (PXD037987). Below is the reviewer's credential for data access.

- Project DOI: 10.6019/PXD037987
- Username: reviewer_pxd037987@ebi.ac.uk
- Password: nloP9oLa.

6. *The MS2 and fold-changes of the important cys sites, such as cys 148 and 284 of IDH3 γ , should be added in the manuscript. Their identification with high-confidence is the foundation of follow-up studies.*

Answer: Though the cys-284 peptide was not identified in this study, cys-148 peptide was successfully measured with high confidence (XCorr values ranging from 2.59-3.63). The fold change for Cys148 was found to be 1.52, p value 0.016. This information is provided in Supplement Table 1 and on page 5, lines 158-160 of the revised manuscript.

7. *The STN is a very weird term in describing fold-change in proteomics. Signal-to-noise has an exact definition in mass spectrometry, but rarely represents fold-change. The authors should revise this word, and related figures and tables.*

Answer: A power law global error model (PLGEM) is a widely accepted method in identifying differentially expressed genes (Pavelka et al., BMC Bioinformatics, 2004 Dec 17;5:203), but indeed is not as frequently utilized in proteomics. We have replaced STN with fold-change to enable easier evaluation by readers.

8. *The overall quality of the redox proteomics should be assessed. In this reviewer's opinion, there are several flaws.*

- a. *Why was the enrichment done on proteins, not on peptides? A peptide-based enrichment would deepen the analysis coverage.*

Answer: The experiment was performed according to the method described in Nat Protoc. 2014 Jan; 9(1): 64–75. The analysis shown by the bar graph above demonstrates significant enrichment and capture of Cys-containing peptides.

- b. *How much was the enrichment efficiency (cys-containing peptides vs. total peptides)? Based on several example raw data the authors provided, this reviewer found that the enrichment efficiency was bad. Roughly, only less than 10% of identified peptides contain cys. This is a sign that the experiments were not performed properly. Moreover, in the several data files I searched, the cys 148 and 284 of IDH3 γ were not identified.*

Answer: To clarify, 3,316 Cys-peptides out of 4,829 total peptides (68%) were found in the original redox proteomics data set submitted with the first version of the manuscript (performed with the Orbitrap Velos Pro). This is higher than the proportion of Cys-peptides found in the global proteomics data (17%). Cys148 peptide was identified in both

the proteomics and redox proteomics data. We indeed did not identify Cys284 in the redox proteomics data. However, the peptide containing Cys284 might not be as susceptible to tryptic cleavage as the Cys148 peptide. The new data set submitted with the revised manuscript (performed with the Orbitrap Eclipse) includes 2,000 more Cys-peptides, i.e. 5391 Cys-containing peptides (see Fig. 3B).

- c. *Total numbers of cys sites and changed cys sites should be added. Their quantitative results should be included.*

Answer: In total we identified 6374 cysteine sites in 5391 peptides. Out of these we found that 185 peptides demonstrated a significant decreased modification and 115 a significant increased modification in the samples obtained from the 7 days D-ala treated HyPer-DAO mice compared to the wt mice. These numbers are now described on page 5, lines 141-148 and Fig. 3B.

- d. *Blocking free cys with MSTP was done in a RIPA buffer, which was a mild buffer thus couldn't denature all proteins' structures. Therefore, some free cys sites might not be blocked in the first step, affecting the subsequent enrichment. The authors should evaluate the completeness of the reaction between free cys and MSTP.*

Answer: We have used RIPA with 2% SDS and higher amount of denaturing agent and also 2% of SDS was used in reconstitution of protein pellet.

9. *Line 107. How does Nrf2 itself change?*

Answer: We checked for NRF2 protein levels using the anti-Nrf2 antibody (Bioworld Technologies, L593). Detecting NRF2 on protein level is challenging (Lau et al, Antioxid Redox Signal. 2013, doi: 10.1089/ars.2012.4754) since some commercially available antibodies detect non-specific bands. Therefore, we used cell extracts of non-treated and H₂O₂ treated C2C12 cells as positive control. In line with the mild induction of Nrf2 target RNA expression, Nrf2 protein levels did not reach the detection level in the protein extracts obtained from left ventricles of wt and HyPer DAO mice compared to the positive control, i.e. cell extract of H₂O₂ treated C2C12 cells.

Nrf2 protein levels in hearts of wild type (wt) and HyPer-DAO transgenic (tg) after treatment with D-ala. Mice were treated with D-ala for 7 days in the drinking water. On day 7 animals were sacrificed and left ventricles were isolated. Protein extracts from the left ventricles were analyzed for Nrf2 and GAPDH protein levels by Western blot. As a positive control for Nrf2, protein extracts of non-treated (n.t.) or H₂O₂ treated C2C12 cells were loaded.

10. Line 169 and fig 3e. The RNA levels of IDH3 were increased, though the protein level was not. Is the difference statistically significant?

Answer: The differences between left ventricles from 7d or 21d wt and HyPer-DAO mice in regard to IDH3 α , β and γ are not statistically significant. Significance was tested by one-sample t-test.

11. Line 171 and fig 3G. Does IDH3 activity really go back to normal after removing D-ala for 2days? Is the difference between the two bars on the right statistically significant?

Answer: The difference between the three groups, i.e. wt-on, HyPer-DAO on and HyPer-DAO on-off was analyzed by Welch`s ANOVA test. The following comparisons were done: wt-on vs. HyPer-DAO on, p-value: 0.0307; wt-on vs. HyPer-DAO off, p-value: 0.8809, HyPer-DAO on vs. HyPer-DAO off, p-value: 0.2652.

12. Fig 7B. the IDH3 basal activity of C236A is lower than wt IDH3 and other mutants. Are there any explanations?

Answer: The basal activity of C236A was not significantly different compared to basal activity of the wt. Still, there seems to be at least a trend towards lower activity as mentioned by the reviewer. This might be explained by inducing changes through mutating a cysteine, that might have no regulatory function in redox modulation but a structural function. A short description and discussion is included on page 8, lines 275-277 of the revised manuscript.

Aside from the experiments performed in response to the reviewer comments, we have in the meantime generated HEK HyPer-DAO-NLS cells that are gene edited on both critical cysteines, i.e. Cys148 and Cys284. The respective C148A/C284A cells were analyzed regarding IDH3 activity and ATP production after D-ala stimulation. In line with our hypothesis, C148/C284A cells did not increase IDH3 activity and did show a decrease in ATP levels in response to the stimulus. The new data set is presented in Suppl. Fig. 5A-C.

We shortened the text throughout the manuscript to meet the required word count (no more than 5000 words in the main text). The word count of the revised abstract is n = 141 and of the revised main text n = 4997.

We would like to thank all reviewers for their comments and hope that with the additional experiments performed and information added, we were able to address all concerns.

REVIEWER COMMENTS

Reviewer #1 (Remarks to the Author):

Authors addressed all the major concerns that were raised. I have a few more comments/suggestions:

1) I take my words back that H₂O₂ levels generated by DAO are modest (Fig. 7H). The DAO-HyPer fusion is indeed a powerful genetically encoded tool for compartment-specific generation of H₂O₂. I anticipate that DAO-HyPer tool will be used in various studies to explore metabolic remodeling due to a pro-oxidative shift. I suggest that authors do INCLUDE as a supplement figure: “The redox cyler menadione but not inhibitors of the ETC induce a ROS response similar to D-ala detected by HyPer2”.

2) I think it needs to be stated somewhere (it will address concerns of Reviewer 2) that because DAO costumes O₂ to make H₂O₂ SeaHorse measurements are not straight forward (and ATP production was used as a proxy for the ETC activity instead [both heart and HEK cells]). This of course has its caveats as in cancer cell lines (HEK and many others) majority of the ATP produced is via glycolysis.

3) I still not quite understand the connection between an increased IDH3 activity as a response to elevated H₂O₂ levels and at the same time a decrease in the TCA cycle activity (authors suggest an INCREASE of flow towards AKG only). It also looks like that succinate levels go up with H₂O₂ regardless of the state of the redox switch in IDH3 (as stated by authors). I think that in Discussion the discovery of the switch should be presented as a major finding as we still do not quite understand the role of this switch and how it is integrated in cellular metabolism. (especially since now authors dialed down on GSH metabolism in the Discussion).

4) I think that data presented in Fig. 7H is truly remarkable. In C148A IDH3 cells there is a drastic increase in NADH/NAD⁺ (a pro-reductive shift). At the same time as expected H₂O₂ production by DAO exhausted NADPH levels (very low NADPH/NADP⁺ and GSH/GSSG ratios). However, in C148A IDH3 there is a sizable increase in NADPH/NADP⁺ ratio with D-Ala (wt vs C148A). This suggests a rebalancing of reducing of NADH and NADPH equivalents (a truly remarkable observation). Also, it is very interesting that inhibition of GSH biosynthesis rescued ATP production in C148A IDH3 cells. On a similar note: it will be interesting to see the GSH/GSSG ratio in C148A IDH3 cells treated with D-Ala.

5) Lines:375-380 The entire paragraph is confusing. How an increase in IDH3 activity (NADH producing enzyme) can lead to NADH decrease? There is an interesting crosstalk between IDH3 as in C148A IDH3 cells the NADH/NAD⁺ ratio was increased substantially (see above)! Again, finding the exact mechanism of this “redox rebalancing” is outside the scope of this work.

6) Minor:

Fig. 5: panels B and C are labeled identically.

Fig. 7H: typo in “NADPH/NADP⁺”

Fig. 7I: change the schematic, it appears that TCA cycle activity is going up with H₂O₂ and wt IDH3.

Reviewer #2 (Remarks to the Author):

The authors have significantly enhanced the manuscript by adding tracer metabolism approaches, and they have properly qualified or removed statements that were not completely supported by the data.

Reviewer #4 (Remarks to the Author):

The authors have addressed all my questions or given reasonable explanations thoughtfully. Thanks.

Reviewer #5 (Remarks to the Author):

All the modifications have been read. Most concerns have been solved, but there is still a problem that I'm concerned about. The clinical relevance of this study remains unclear. Authors have demonstrated the changes of IDH3 γ induced by H₂O₂ also existed in TAC model. However, authors failed to demonstrate that if inhibiting IDH3 γ Cys148 modification could suppress the progression of cardiac dysfunction and fibrosis in TAC model in vivo, which is closely connected with the clinical significance of this study.

Reviewer comments - Point to Point response

We would like to thank all reviewers for their input.

A more detailed point to point response can be seen below:

Reviewer #1

Authors addressed all the major concerns that were raised.

We would like to thank reviewer #1 for this statement.

I have a few more comments/suggestions:

1) *I take my words back that H₂O₂ levels generated by DAO are modest (Fig. 7H). The DAO-HyPer fusion is indeed a powerful genetically encoded tool for compartment-specific generation of H₂O₂. I anticipate that DAO-HyPer tool will be used in various studies to explore metabolic remodeling due to a pro-oxidative shift. I suggest that authors do INCLUDE as a supplement figure: "The redox cyclers menadione but not inhibitors of the ETC induce a ROS response similar to D-ala detected by HyPer2".*

As suggested, we have now included the data set obtained from the stimulation experiments using menadione and the inhibitors of the ETC to see the ROS response in comparison to the D-ala stimulation (see Supplementary Figure 2A and lines 177-180).

2) *I think it needs to be stated somewhere (it will address concerns of Reviewer 2) that because DAO consumes O₂ to make H₂O₂ seahorse measurements are not straight forward (and ATP production was used as a proxy for the ETC activity instead [both heart and HEK cells]). This of course has its caveats as in cancer cell lines (HEK and many others) majority of the ATP produced is via glycolysis.*

A respective statement has been included in the revised manuscript (see lines 218-219).

3) *I still not quite understand the connection between an increased IDH3 activity as a response to elevated H₂O₂ levels and at the same time a decrease in the TCA cycle activity (authors suggest an INCREASE of flow towards AKG only). It also looks like that succinate levels go up with H₂O₂ regardless of the state of the redox switch in IDH3 (as stated by authors). I think that in Discussion the discovery of the switch should be presented as a major finding as we still do not quite understand the role of this switch and how it is integrated in cellular metabolism. (especially since now authors dialed down on GSH metabolism in the Discussion).*

We certainly agree with this statement. We have pointed out in the revised manuscript that the discovery of IDH3 γ as a redox switch is the major finding in the abstract (see line 42) as well as in the discussion (see lines 394-396).

4) *I think that data presented in Fig. 7H is truly remarkable. In C148A IDH3 cells there is a drastic increase in NADH/NAD⁺ (a pro-reductive shift). At the same time as expected H₂O₂ production by DAO exhausted NADPH levels (very low NADPH/NADP⁺ and GSH/GSSG ratios). However, in C148A IDH3 there is a sizable increase in NADPH/NADP⁺ ratio with D-Ala (wt vs C148A). This suggests a rebalancing of reducing of NADH and NADPH equivalents (a truly remarkable observation). Also, it is very interesting that inhibition of GSH biosynthesis rescued ATP production in C148A IDH3 cells. On a similar note: it will be interesting to see the GSH/GSSG ratio in C148A IDH3 cells treated with D-Ala.*

We would like to thank the reviewer for this positive statement. As suggested we have included GSH and GSSG measurements obtained from wt versus C148A IDH3 cells (see Fig. 7I and lines 378-380).

5) *Lines:375-380 The entire paragraph is confusing. How an increase in IDH3 activity (NADH producing enzyme) can lead to NADH decrease? There is an interesting crosstalk between IDH3 as in*

C148A IDH3 cells the NADH/NAD+ ratio was increased substantially (see above)! Again, finding the exact mechanism of this “redox rebalancing” is outside the scope of this work.

We apologize for this confusion. The reviewer is right by mentioning that the NADH/NAD+ ratio was significantly increasing in the sense of a pro-reductive shift. We have changed the wording of the mentioned paragraph accordingly (see lines 378-380).

6) *Minor:*

Fig. 5: panels B and C are labeled identically.

We have changed the labeling to clearly indicate that Fig. 5 panel B and C show ATP measurements in HEK cells and isolated cardiomyocytes, respectively.

Fig. 7H: typo in “NADPH/NADP+”

The typo has been corrected.

Fig. 7I: change the schematic, it appears that TCA cycle activity is going up with H₂O₂ and wt IDH3.

We have changed the schematic to make this point clearer.

Reviewer #2 *The authors have significantly enhanced the manuscript by adding tracer metabolism approaches, and they have properly qualified or removed statements that were not completely supported by the data.*

We would like to thank reviewer #2 for this positive statement.

Reviewer #4

The authors have addressed all my questions or given reasonable explanations thoughtfully. Thanks.

We would like to thank reviewer #4 for this positive statement.

Reviewer #5

All the modifications have been read. Most concerns have been solved, but there is still a problem that I'm concerned about. The clinical relevance of this study remains unclear. Authors have demonstrated the changes of IDH3 γ induced by H₂O₂ also existed in TAC model. However, authors failed to demonstrate that if inhibiting IDH3 γ Cys148 modification could suppress the progression of cardiac dysfunction and fibrosis in TAC model in vivo, which is closely connected with the clinical significance of this study.

In order to prove the importance of IDH3 γ Cys148 modification fully, a Cys148 gene edited mouse model would be necessarily needed to be established. It is uncertain, if a respective mouse model would be viable. As suggested by the editor we have therefore addressed this point by clearly stating this limitation in the text (see lines 369-371).

REVIEWERS' COMMENTS

Reviewer #1 (Remarks to the Author):

The authors have addressed all my questions and reasonable explanations were given. Thank you.

Very minor: Fig. 7I , not clear statistics of which columns is shown (shifted label maybe?)

Reviewer comments - Point to Point response

We would like to thank all reviewers for their input.

A more detailed point to point response can be seen below:

Reviewer #1 (Remarks to the Author):

The authors have addressed all my questions and reasonable explanations were given. Thank you.

We would like to thank reviewer #1 for this positive feedback.

Very minor: Fig. 7I, not clear statistics of which columns is shown (shifted label maybe?)

We have checked Fig. 7I and have aligned the connecting lines between the columns to show statistically significant differences in a clear manner.